# Minimizing Upper Confidence Bounds: A Data-Driven Framework for Stochastic Programming

**Shixin Liu** [1]   **Ming Gao** [1]   **Jian Hu** [1]

## Abstract

Stochastic programming is often challenged by epistemic uncertainty, where critical probability distributions are poorly characterized or unknown due to a lack of data. To address this, we pioneer a novel framework for stochastic programming that minimizes an upper confidence bound (UCB) on the expected random cost, acting as a robustness-seeking strategy. Our central contribution is the Average Percentile Upper Bound (APUB), a new statistical construct that serves as both a statistically rigorous upper bound for population means and an approximate risk metric for sample means. We rigorously prove the asymptotic correctness and consistency of APUB, establishing a reliable foundation for data-driven decision-making. We also develop practical solution methods, including a bootstrap sampling approximation method and an L-shaped method, to solve APUB optimization problems, with a specific focus on two-stage linear stochastic optimization with random recourse. Empirical demonstrations on a two-stage product mix problem reveal the significant benefits of our APUB optimization framework, which fortifies the process against epistemic uncertainty while reinforcing key decision-making attributes like reliability and consistency. The implementation and source code are available at https://github.com/8Wings/APUB-Optimization.

## 1. Introduction

Optimizing decisions under uncertainty is central to modern decision-making. Classical stochastic programming provides a principled framework, but it typically assumes that the probability distribution governing uncertain parameters is known. In practice, the distribution is rarely available with full accuracy, and ignoring this *epistemic uncertainty*—uncertainty due to limited data or imperfect modeling—can yield decisions that are overly optimistic and fragile when deployed.

To mitigate epistemic uncertainty, the optimization literature has developed robust and distributionally robust optimization (DRO) frameworks that seek solutions performing well under distributional ambiguity. In this paper, we take a complementary route: rather than optimizing under an explicit ambiguity set, we integrate *statistical inference* directly into stochastic optimization through *upper confidence bound (UCB) minimization*. The key idea is to replace the unknown objective value with an upper confidence bound constructed from data, thereby selecting decisions that remain reliable under estimation error.

Consider the expectation minimization problem

$$\min_{x \in \mathcal{X}} \mathbb{E}[F(x, \xi)], \qquad \text{(EM-M)}$$

where $\mathcal{X}$ is the feasible set, $\xi$ follows an unknown distribution $\mathbb{P}$, and $F$ is a cost function. Given an i.i.d. sample of size $N$ and its empirical distribution $\widehat{\mathbb{P}}_N$, the Sample Average Approximation (SAA) model

$$\min_{x \in \mathcal{X}} \mathbb{E}\left[ F(x, \xi) \,\Big|\, \widehat{\mathbb{P}}_N \right], \qquad \text{(SAA-M)}$$

is a natural estimator of (EM-M). While effective for large $N$, SAA can be biased and unstable in small-sample regimes. We therefore propose to optimize a $100(1 - \alpha)\%$ UCB for $\mathbb{E}[F(x, \xi)]$ constructed from $\widehat{\mathbb{P}}_N$. Denoting this bound by $\mathbb{U}^\alpha\left[ \mathbb{E}[F(x, \xi)] \mid \widehat{\mathbb{P}}_N \right]$, we study the UCB optimization model

$$\min_{x \in \mathcal{X}} \mathbb{U}^\alpha \left[ \mathbb{E}[F(x, \xi)] \,\Big|\, \widehat{\mathbb{P}}_N \right]. \qquad \text{(UCB-M)}$$

Model (UCB-M) seeks decisions that are not only good under the point estimate but also robust to sampling variability by controlling an upper confidence guarantee on the expected cost.

[1]Department of Industrial & Manufacturing System Engineering, University of Michigan-Dearborn, 4901 Evergreen Rd.,Dearborn, 48128, MI, USA. Correspondence to: Jian Hu <jianhu@umich.edu>.

*Proceedings of the 43rd International Conference on Machine Learning*, Seoul, South Korea. PMLR 306, 2026. Copyright 2026 by the author(s).

## 1.1. Related Work

**DRO and UCB optimization.** Epistemic uncertainty in stochastic optimization is commonly addressed via distributionally robust optimization (DRO) (Rahimian & Mehrotra, 2019; Lin et al., 2022). Unlike Bayesian methods that typically assume a parametric model, DRO seeks decisions that perform well over a set of plausible data-generating distributions rather than a single estimated distribution. Two canonical ambiguity-set constructions are: (a) moment-based sets that constrain low-order moments (Calafiore & Ghaoui, 2006; Delage & Ye, 2010; Wiesemann et al., 2014), and (b) discrepancy-based sets defined by a statistical distance to a nominal distribution, e.g., $\phi$-divergences (Read & Cressie, 1988; Ben-Tal et al., 2013; Bayraksan & Love, 2015) and Wasserstein balls (Mohajerin Esfahani & Kuhn, 2018; Blanchet & Murthy, 2019; Xie, 2020; Duque et al., 2022; Gao & Kleywegt, 2023). Wasserstein DRO is especially popular due to its finite-sample guarantees and asymptotic consistency.

In parallel, UCB-style methods incorporate inference directly into the optimization objective. UCB maximization is a standard tool for exploration in reinforcement learning, including multi-armed bandits (Slivkins, 2019) and sample-efficient variants of Q-learning (Wang et al., 2020), and is also used in Bayesian optimization as a principled acquisition rule (Fan et al., 2024). Our setting differs: rather than using UCBs to select informative actions, we use *UCB minimization* as a robustness-seeking criterion in stochastic programming. Specifically, the model in UCB-M selects decisions by minimizing an upper confidence bound of the objective, targeting solutions that are less sensitive to distributional/estimation uncertainty and thus better aligned with worst-case performance under limited data.

**Statistical confidence interval.** Classical one- and two-sided confidence intervals have well-understood large-sample properties. In particular, first-order accuracy corresponds to coverage error $O(N^{-1/2})$, while second-order accuracy corresponds to $O(N^{-1})$ (van der Vaart, 1998). This asymptotic calibration, together with a standardized interpretation of the nominal level, makes statistical upper bounds attractive building blocks for robust-yet-interpretable optimization objectives. A standard frequentist construction uses the sample mean with a normal approximation (often justified for moderate-to-large $N$ via the CLT) (Devore, 2009; Hazra, 2017). Bootstrap methods provide nonparametric alternatives: Efron's percentile interval uses quantiles of the bootstrap sampling distribution (Efron, 1981) and is typically first-order accurate, while the $\text{BC}_a$ interval can achieve second-order accuracy through bias and acceleration corrections (Efron, 1987). In machine learning, upper confidence bounds are frequently derived from concentration inequalities, including Hoeffding-type bounds (Auer et al., 2002),

empirical Bernstein bounds that adapt to sample variance (Mnih et al., 2008), and self-normalized bounds designed for sequential/auto-correlated data (Abbasi-Yadkori et al., 2011). While broadly applicable, such bounds can be difficult to optimize (e.g., non-smooth/non-convex) or overly conservative, motivating data-driven UCB constructions that better balance statistical validity and computational tractability in optimization.

## 1.2. Major Contributions

(i) *UCB minimization for stochastic programming.* We propose a novel UCB minimization framework, UCB-M, to systematically address epistemic uncertainty in stochastic programming. Unlike the UCB maximization typically employed for exploration in reinforcement learning, our criterion is robustness-seeking. By minimizing a UCB on the expected cost, the framework favors decisions that are resilient to estimation errors and aligned with reliable performance under data scarcity. (ii) *Average Percentile Upper Bound (APUB).* We introduce the APUB, a novel statistic that serves a dual role as both a UCB for the population mean and an approximate risk measure for the sample mean. After establishing its asymptotic correctness and consistency, we embed it into UCB-M to create a data-driven model with parameter transparency. There are two major advantages as follows. a) Statistical Interpretation: Unlike DRO, where selecting a statistically meaningful uncertainty radius remains a significant challenge, APUB's uncertainty knob is directly tied to a frequentist confidence level $1 - \alpha$ (e.g., 95%). b) Automatic Convergence: APUB achieves asymptotic convergence for any fixed $\alpha$, bypassing the manual, case-specific tuning required in DRO—where radii must often decay at specific, non-trivial rates to avoid extreme over-conservatism or loss of consistency. APUB instead leverages bootstrap distribution contraction to naturally adapt as the sample size increases. (iii) *Efficient algorithms via bootstrap sampling approximation.* We develop a practical solution method for APUB optimization using bootstrap sampling, specifically tailored for two-stage linear models with random recourse. In this challenging setting, DRO often lacks a tractable dual reformulation, leading to NP-hard inner maximization. Our approach extends the L-shaped method and leverages a unique structural property of the APUB to accelerate second-stage computations via sorting. This enables the framework to scale efficiently even with large bootstrap ensembles.

## 2. Average Percentile Upper Bound

We establish formal definitions for the key concepts employed in this paper. Consider an induced probability space $(\Xi, \mathfrak{B}, \mathbb{P})$, where $\Xi \subseteq \mathbb{R}^{d_\xi}$ is the support of a random vector, $\mathfrak{B}$ is the Borel $\sigma$-algebra, and $\mathbb{P}$ is a proba-

bility measure. Let $(\xi_1, \ldots, \xi_N) \sim \mathbb{P}$ indicate an independent and identically distributed (i.i.d) random sample with a size of $N$ generated from $(\Xi, \mathfrak{B}, \mathbb{P})$. The empirical distribution associated with the random sample is represented as $\widehat{\mathbb{P}}_N := \frac{1}{N} \sum_{n=1}^{N} \delta_{\xi_n}$, where $\delta_{\xi_n}$ is the Dirac delta function at $\xi_n$. As $N$ increases to infinity, we have a sample path $(\xi_1, \xi_2, \ldots)$. Without loss of generality, we ignore the decision variable $x$ and focus our discussion on a measurable cost function $F : \Xi \to \mathbb{R}$ in this section. Denote by $\mu := \mathbb{E}[F(\xi)]$ the population mean and by $\sigma^2 := \mathbb{E}[(F(\xi) - \mu)^2]$ the population variance. We assume $\mu$ and $\sigma$ to be finite in the subsequent analysis. Let $\widehat{\mu}_N := \mathbb{E}[F(\zeta)|\widehat{\mathbb{P}}_N]$ be the sample mean and $\widehat{\sigma}_N^2 := \mathbb{E}[(F(\zeta) - \widehat{\mu}_N)^2|\widehat{\mathbb{P}}_N]$ be the asymptotic sample variance, where $\zeta \sim \widehat{\mathbb{P}}_N$.

Using the bootstrap percentile method, Efron (1981) presents a $100(1 - \alpha)\%$ bootstrap-based UCB for $\mu$ as

$$\mathbb{U}_{\text{Efron}}^{\alpha}[\mu|\widehat{\mathbb{P}}_N] :=$$

$$\inf \left\{ t \in \mathbb{R} : \Pr\left( \frac{1}{N} \sum_{n=1}^{N} F(\zeta_n) \leq t \,\middle|\, \widehat{\mathbb{P}}_N \right) \geq 1 - \alpha \right\}.$$

While this percentile-based UCB is satisfactory in many statistical analyses, it is non-convex and difficult to control/optimize for highly skewed distributions, which are regarded as inferior properties in the realm of optimization. Therefore, we extend Efron's UCB by averaging over the values to the right of the $100(1 - \alpha)$-th percentile.

**Definition 2.1.** The *average percentile* upper bound for $\mu$ with a nominal level $(1 - \alpha)$ is denoted as

$$\mathbb{U}_{\text{APUB}}^{\alpha}[\mu|\widehat{\mathbb{P}}_N] := \frac{1}{\alpha} \int_0^{\alpha} \mathbb{U}_{\text{Efron}}^{\tau}[\mu|\widehat{\mathbb{P}}_N] d\tau. \qquad \text{(APUB)}$$

We can interpret Efron's UCB, alternatively in the realm of risk management and decision making, as an approximation of the Value at Risk (VaR) of $\widehat{\mu}_N$ by substituting $\widehat{\mathbb{P}}_N$ for the true distribution $\mathbb{P}$ of the random variable $\xi_n$ in $\text{VaR}_{\alpha}(\widehat{\mu}_N) = \inf\left\{ t \in \mathbb{R} : \Pr\left( \frac{1}{N} \sum_{n=1}^{N} F(\xi_n) \leq t \right) \geq 1 - \alpha \right\}$. Analogously, APUB approximates the Conditional Value at Risk (CVaR) of $\widehat{\mu}_N$, $\text{CVaR}_{\alpha}(\widehat{\mu}_N) = \frac{1}{\alpha} \int_0^{\alpha} \text{VaR}_{\tau}(\widehat{\mu}_N) d\tau$. APUB serves a dual purpose: as an upper bound for the population mean in statistics and as an approximate risk measure for the sample mean in risk assessment. As a risk measure, it primarily focuses on approximating the tail distribution of the potential estimation error of the population mean, which could result from an inadequacy of sample points. Furthermore, APUB complies with fundamental properties of a coherent risk measure, such as sub-additivity, homogeneity, convexity, translational invariance, and monotonicity. These characteristics make APUB a good candidate to be applied

to stochastic optimization under epistemic uncertainty, particularly in scenarios requiring solvability, such as two-stage stochastic optimization with random recourse. Analogous to Theorem 1 in (Rockafellar & Uryasev, 2000), the following proposition provides an alternative representation for APUB.

**Proposition 2.2.**

$$\mathbb{U}_{\text{APUB}}^{\alpha}[\mu|\widehat{\mathbb{P}}_N]$$

$$= \inf_{t \in \mathbb{R}} \left\{ t + \frac{1}{\alpha} \mathbb{E}\left[ \left[ \frac{1}{N} \sum_{n=1}^{N} F(\zeta_n) - t \right]_+ \,\middle|\, \widehat{\mathbb{P}}_N \right] \right\}$$

$$= \inf_{t \in \mathbb{R}} \left\{ t + \frac{1}{\alpha} \int \left[ \frac{1}{N} \sum_{n=1}^{N} F(\zeta_n) - t \right]_+ \prod_{n=1}^{N} \widehat{\mathbb{P}}_N(d\zeta_n) \right\},$$

*where the bold integral symbol means an $N$-fold integral over the $N$-fold product of $\widehat{\mathbb{P}}_N$.*

*Remark* 2.3. By Proposition 2.2, we have that $\mathbb{U}_{\text{APUB}}^{\alpha}[\mu|\widehat{\mathbb{P}}_N]$ monotonically decreases in $\alpha \in (0, 1]$ w.p.1. This implies that, for $\alpha \in (0, 1]$,

$$\mathbb{U}_{\text{APUB}}^{\alpha}[\mu|\widehat{\mathbb{P}}_N] \geq \mathbb{U}_{\text{APUB}}^{1}[\mu|\widehat{\mathbb{P}}_N] = \mathbb{E}\left[ \frac{1}{N} \sum_{n=1}^{N} F(\zeta_n) \,\middle|\, \widehat{\mathbb{P}}_N \right]$$

$$= \frac{1}{N} \sum_{n=1}^{N} \mathbb{E}\left[ F(\zeta_n) \,\middle|\, \widehat{\mathbb{P}}_N \right] = \widehat{\mu}_N, \quad \text{w.p.1.}$$

**Proposition 2.4.** *Suppose that the skewness $\mathbb{E}[F(\xi) - \mu]^3/\sigma^3 < \infty$. Then, for a fixed nominal level $1 - \alpha$, $\mathbb{U}_{\text{APUB}}^{\alpha}[\mu|\widehat{\mathbb{P}}_N]$ is first-order asymptotically correct, i.e.,*

$$\Pr\left( \mu \leq \mathbb{U}_{\text{APUB}}^{\alpha}[\mu|\widehat{\mathbb{P}}_N] \right) \geq (1 - \alpha) + O(N^{-1/2}).$$

Proposition 2.4 establishes the asymptotic correctness of APUB, which guarantees that the nominal level is a conservative boundary for the actual coverage probability. This attribute confirms that APUB is an effective upper-bound statistic, especially valuable for its robust response to epistemic uncertainty encountered with limited sample data.

**Theorem 2.5.** *For any $\alpha \in (0, 1]$, as $N \to \infty$, $\mathbb{U}_{\text{APUB}}^{\alpha}[\mu|\widehat{\mathbb{P}}_N] \to \mu$, w.p.1.*

Theorem 2.5 shows the asymptotic consistency, i.e. APUB is a consistent estimator for the population mean, given that the uncertainty diminishes along with an increase in the sample size,

## 3. APUB Optimization

We apply APUB to stochastic optimization problems. In the context of optimization, we have a decision region $\mathcal{X} \subseteq \mathbb{R}^{d_x}$ and let the cost function $F(x, \xi) : \mathcal{X} \times \Xi \mapsto \mathbb{R}$ be $\mathfrak{B}$-measurable for all $x \in \mathcal{X}$. Denote the mean and standard

deviation of $F(x, \xi)$ by $\mu(x)$ and $\sigma(x)$ respectively. The UCB-M framework using APUB is written as

$$\min_{x \in \mathcal{X}} \mathbb{U}_{\text{APUB}}^{\alpha}[\mu(x)|\widehat{\mathbb{P}}_N]. \qquad \text{(APUB-M)}$$

Let $\widehat{\vartheta}_N^{\alpha}$ be the optimal value of APUB-M and $\widehat{\mathcal{S}}_N^{\alpha}$ denote the set of its optimal solutions. Also, denote by $\vartheta^*$ the optimal value of EM-M and by $\mathcal{S}$ the set of its optimal solutions. By Remark 2.3, we know that $\widehat{\vartheta}_N^{\alpha}$ decreases in $\alpha \in (0, 1]$ w.p.1 and $\widehat{\vartheta}_N^1$ is the optimal value of SAA-M.

By applying the statistical technique, we expect to significantly reduce the impact of epistemic uncertainty. We now present some mild assumptions as follows.

**Assumption 3.1.** There exists a compact set $\mathcal{K} \subseteq \mathcal{X}$ such that: (i) $\mathcal{S} \subseteq \mathcal{K}$; (ii) $\widehat{\mathcal{S}}_N^{\alpha} \subseteq \mathcal{K}$ w.p.1 for sufficiently large $N$ and $\alpha \in (0, 1]$.

Assumption 3.1 is frequently encountered in the literature pertaining to the asymptotic analysis of the SAA method (Birge & Louveaux, 2011; Shapiro et al., 2021). This assumption posits that it is adequate to confine the examination of decision properties to the compact set $\mathcal{K}$. For the purposes of the discussion in the remainder of Section 3, we proceed under the premise that the decision space is indeed $\mathcal{K}$, a simplification that does not limit the generality of our analysis.

**Assumption 3.2.** There exists an open convex hull $\mathcal{N}$ containing $\mathcal{K}$ such that: (i) $F(x, \xi)$ is convex on $\mathcal{N}$ for each $\xi \in \Xi$; (ii)$\mu(x)$ and $\sigma(x)$ are finite for all $x \in \mathcal{N}$.

**Asymptotic Correctness:** Mohajerin Esfahani & Kuhn (2018) introduce the concept of reliability for a certain optimal solution in DRO approaches. The reliability refers to the probability that the optimal value of a DRO model exceeds the expected cost of the system at its optimal solution in true scenarios. We extend this concept to the entire optimal solution set, which in our case is called the coverage probability of the general UCB-M framework. Denote a probability function of a given subset $S \subseteq \mathcal{X}$ as

$$\beta(\vartheta, S) := \Pr\left(\vartheta \geq \max_{x \in S} \mu(x)\right). \qquad (1)$$

Let $\overline{\vartheta}_N^{\alpha}$ and $\overline{\mathcal{S}}_N^{\alpha}$ be the optimal value and optimal solution set of UCB-M, respectively. The coverage probability of UCB-M is $\beta(\overline{\vartheta}_N^{\alpha}, \overline{\mathcal{S}}_N^{\alpha})$, which measures the chance that $\overline{\vartheta}_N^{\alpha}$ can serve as an upper bound of the actual performance of UCB-M across all optimal solutions. In the following, we define the asymptotic correctness of UCB-M.

**Definition 3.3.** UCB-M is asymptotically correct if the limit of its coverage probability serves as a lower bound for the nominal level as $\lim_{N \to \infty} \beta(\overline{\vartheta}_N^{\alpha}, \overline{\mathcal{S}}_N^{\alpha}) \geq (1 - \alpha)$.

Defined on the entire optimal solution set, the concept of asymptotic correctness is stricter than the reliability concerning a certain optimal solution. If a UCB-M framework is asymptotically correct, we have that, for any $\bar{x}_N \in \overline{\mathcal{S}}_N^{\alpha}$,

$$\lim_{N \to \infty} \beta(\overline{\vartheta}_N^{\alpha}, \{\bar{x}_N\}) \geq \lim_{N \to \infty} \beta(\overline{\vartheta}_N^{\alpha}, \overline{\mathcal{S}}_N^{\alpha}) \geq (1 - \alpha).$$

In the subsequent statement, we refer to $\beta(\overline{\vartheta}_N^{\alpha}, \{\bar{x}_N\})$ as the coverage probability of UCB-M concerning $\bar{x}$, or simply the coverage probability at $\bar{x}_N$. The coverage probability quantifies the likelihood that $\overline{\vartheta}_N^{\alpha}$ is an upper bound for the true expected cost at $\bar{x}_N$, thereby indicating the reliability of the cost estimation for an obtained solution. Thus, we can say that the asymptotic correctness of UCB-M guarantees the asymptotic correctness at any optimal solution. The following theorem shows the asymptotic correctness of APUB-M.

**Theorem 3.4.** *Suppose that Assumption 3.1 and 3.2 hold. Assume that (i) there exists $x_0 \in \mathcal{K}$ such that $\mathbb{E}\left[|F(x_0, \xi)|^3\right] < \infty$; (ii) for any $x, y \in \mathcal{K}$, $|F(x, \xi) - F(y, \xi)| < L(\xi)\|x - y\|$, where $\mathbb{E}\left[|L(\xi)|^3\right] < \infty$; and (iii) $\sigma(x)$ and $\sigma^{-1}(x)$ are bounded on $x \in \mathcal{K}$. Then, APUB-M is asymptotically correct for $\alpha \in (0, 1]$, i.e.,*

$$\lim_{N \to \infty} \beta(\widehat{\vartheta}_N^{\alpha}, \widehat{\mathcal{S}}_N^{\alpha}) \geq (1 - \alpha).$$

*Remark* 3.5. The attribute of asymptotic correctness lends APUB-M interpretability in the context of statistics. This means that the decision-maker can intuitively set the desired reliability level of APUB-M by selecting an appropriate nominal level $(1 - \alpha)$. This direct statistical interpretation does not require detailed physical insights into the problems being addressed. Section 5 provides a numerical demonstration of how this model interpretability confers an advantage.

*Remark* 3.6. Recall $\vartheta^*$ is the optimal value of EM-M. Since $\vartheta^* \leq \mu(x)$ for all $x \in \widehat{\mathcal{S}}_N^{\alpha}$, we obtain that

$$\lim_{N \to \infty} \Pr\left(\widehat{\vartheta}_N^{\alpha} \geq \vartheta^*\right) \geq \lim_{N \to \infty} \beta(\widehat{\vartheta}_N^{\alpha}, \widehat{\mathcal{S}}_N^{\alpha}) \geq (1 - \alpha).$$

This implies that the nominal level approximately represents the lower bound of the probability that $\widehat{\vartheta}_N^{\alpha}$ serves as an upper bound for $\vartheta^*$.

**Asymptotic Consistency:** In optimization, asymptotic consistency refers to the convergence of the optimal value and optimal solution set of APUB-M with their counterparts in EM-M w.p.1 as the sample size increases. The following theorem exhibits the asymptotic consistency of APUB-M.

**Theorem 3.7.** *Suppose Assumptions 3.1 and 3.2 hold. Then, for any given $\alpha \in (0, 1]$, as $N \to \infty$, we have $\widehat{\vartheta}_N^{\alpha} \to \vartheta^*$ and $\mathbb{D}(\widehat{\mathcal{S}}_N^{\alpha}, \mathcal{S}) := \sup_{y \in \widehat{\mathcal{S}}_N^{\alpha}} \inf_{z \in \mathcal{S}} \|y - z\| \to 0$ w.p.1.*

*Remark* 3.8. A key advantage of APUB-M is its consistency and data-driven nature, as sample size is the sole

factor that determines both convergence and conservatism. As epistemic uncertainty diminishes with a larger sample size, APUB naturally contracts. This makes APUB-M less conservative, all without requiring any additional tuning or parameter rescaling. Our approach calibrates reliability with the nominal level $(1 - \alpha)$ but does not need to specify or rescale it to avoid over-conservatism as more data become available.

# 4. Solution Method Based on Bootstrap Sampling Approximation

By Proposition 2.2, problem (APUB-M) can be written as

$$\min_{(x,t)\in\mathcal{X}\times\mathbb{R}} t + \frac{1}{\alpha} \int \Big[\frac{1}{N}\sum_{n=1}^{N} F(x,\zeta_n) - t\Big]_+ \prod_{n=1}^{N} \widehat{\mathbb{P}}_N(d\zeta_n). \tag{2}$$

Recall that $\widehat{\mathbb{P}}_N$ is the empirical distribution associated with the original sample $(\xi_1, \ldots, \xi_N)$. A bootstrap sample $(\zeta_1, \ldots, \zeta_N)$ drawn from $\widehat{\mathbb{P}}_N$ can be represented by the multiplicities $V_n := \big|\{j : \zeta_j = \xi_n\}\big|$. By construction, $V_n \geq 0$ and $\sum_{n=1}^{N} V_n = N$, and $V := (V_1, \ldots, V_N)$ follows a multinomial distribution with index $N$ and parameter vector $(1/N, \ldots, 1/N)$. Let $\mathfrak{M}_N(v)$ denote its probability mass function and define $\mathfrak{V} := \Big\{v \in \mathbb{Z}_+^N : \sum_{n=1}^{N} v_n = N\Big\}$. Then (2) is equivalent to

$$\min_{(x,t)\in\mathcal{X}\times\mathbb{R}} t + \frac{1}{\alpha} \sum_{v\in\mathfrak{V}} \Big[\frac{1}{N}\sum_{n=1}^{N} v_n F(x,\xi_n) - t\Big]_+ \mathfrak{M}_N(v). \tag{3}$$

The support $\mathfrak{V}$ contains $|\mathfrak{V}| = \binom{2N-1}{N}$ scenarios, so directly solving (3) is intractable for moderate $N$.

To mitigate this combinatorial growth, we approximate the expectation in (3) via Monte Carlo sampling. Let $(V_{m,1}, \ldots, V_{m,N}) \sim \mathfrak{M}_N(v)$ for $m = 1, \ldots, M$. Replacing the expectation by a sample average yields the bootstrap SAA problem

$$\min_{(x,t)\in\mathcal{X}\times\mathbb{R}} t + \frac{1}{\alpha M} \sum_{m=1}^{M} \Big[\frac{1}{N}\sum_{n=1}^{N} V_{m,n} F(x,\xi_n) - t\Big]_+. \tag{4}$$

In practice, a much smaller number of bootstrap scenarios, $M \ll \binom{2N-1}{N}$, already provides an accurate approximation. The asymptotic properties of such SAA schemes, including convergence and rates under mild assumptions, are well documented; see, e.g., Shapiro et al. (2021).

## 4.1. Two-Stage Linear Stochastic Programs with Random Recourse

We now specialize APUB to a two-stage linear stochastic program with random recourse. A bootstrap approximation

(4) of the first stage is formulated as

$$\begin{aligned}
\min_{x,t} \quad & c^\top x + t + \frac{1}{\alpha M} \sum_{m=1}^{M} \Big[\frac{1}{N}\sum_{n=1}^{N} V_{m,n} Q(x,\xi_n) - t\Big]_+ \\
\text{s.t.} \quad & Ax = b, \\
& x \geq 0,
\end{aligned} \tag{5}$$

where $\xi_n = (q_n, h_n, T_n, W_n), n = 1, \ldots, N$, are the observations, and $Q$ is a recourse cost function as

$$\begin{aligned}
Q(x,\xi_n) := \min_{y} \quad & q_n^\top y \\
\text{s.t.} \quad & W_n y = h_n - T_n x, \\
& y \geq 0.
\end{aligned} \tag{6}$$

## 4.2. Adapted L-Shaped Method (see Appendix A)

We adapt the L-shaped method (Van Slyke & Wets, 1969; Birge & Louveaux, 2011) to solve solve (5).

**Master problem.** At iteration $(\ell, k)$ we consider the master problem

$$\begin{aligned}
\min_{x,\eta} \quad & c^\top x + \eta & & \text{(7a)} \\
\text{s.t.} \quad & Ax = b, & & \text{(7b)} \\
& D_j x \geq d_j, & j = 1, \ldots, \ell, & \text{(7c)} \\
& E_j x + \eta \geq e_j, & j = 1, \ldots, k, & \text{(7d)} \\
& x \geq 0, & & \text{(7e)}
\end{aligned}$$

where (7c) and (7d) collect feasibility and optimality cuts, respectively. Let $(\hat{x}, \hat{\eta})$ be an optimal master solution. If $k = 0$ (no optimality cuts), we set $\hat{\eta} = -\infty$ and ignore it when solving (7).

**Feasibility cuts.** For a given $\hat{x}$, the second-stage problem (6) with data $\xi_n$ may be infeasible, in which case $Q(\hat{x}, \xi_n) = +\infty$. Define

$$\gamma(x) := \min_{t\in\mathbb{R}} \Big\{t + \frac{1}{\alpha M} \sum_{m=1}^{M} \big[r_m(x) - t\big]_+\Big\}, \tag{8}$$

where

$$r_m(x) := \frac{1}{N} \sum_{n=1}^{N} V_{m,n} Q(x,\xi_n), \quad m = 1, \ldots, M. \tag{9}$$

Feasibility is checked by solving, for each $n = 1, \ldots, N$,

$$\begin{aligned}
u_n(\hat{x}) := \min_{y,v^+,v^-} \quad & \mathbf{1}^\top v^+ + \mathbf{1}^\top v^- & & \text{(10)} \\
\text{s.t.} \quad & W_n y + I v^+ - I v^- = h_n - T_n \hat{x}, \\
& y \geq 0, \ v^+ \geq 0, \ v^- \geq 0,
\end{aligned}$$

where $\mathbf{1}$ is the all-ones vector and $I$ is the identity matrix. If $u_n(\hat{x}) > 0$ for some $n$, then $Q(\hat{x}, \xi_n) = +\infty$. Let $\phi_n$

be the simplex multipliers associated with (10); we add the feasibility cut

$$D_{\ell+1}x \geq d_{\ell+1}, \quad D_{\ell+1} := \phi_n^\top T_n, \quad d_{\ell+1} := \phi_n^\top h_n,$$

increment $\ell$, and resolve the master problem.

**Optimality cuts.** If $u_n(\hat{x}) = 0$ for all $n$, the second-stage problems are feasible and we generate an optimality cut. We solve $Q(\hat{x}, \xi_n)$ for each $n$. let $\psi_n$ be the simplex multipliers associated with (6). We compute $r_m(\hat{x})$ via (9) and sort the values in ascending order: $r_{(1)}(\hat{x}) \leq \cdots \leq r_{(M)}(\hat{x})$, where the index $(\cdot)$ denotes the permutation induced by the sort. We use the same indexing for the bootstrap weights, i.e., $V_{(m),n}$ is the $n$-th component of the bootstrap vector associated with $r_{(m)}(\hat{x})$. Let $J := \lceil (1-\alpha)M \rceil$ and define

$$\hat{\lambda} := \left(1 - \frac{M-J}{\alpha M}\right) r_{(J)}(\hat{x}) + \frac{1}{\alpha M} \sum_{m=J+1}^{M} r_{(m)}(\hat{x}). \tag{11}$$

If $\hat{\eta} \geq \hat{\lambda}$, then $\hat{x}$ is optimal for (5) and the algorithm terminates. Otherwise, we form the coefficients of a new optimality cut as

$$E_{k+1} := \left(1 - \frac{M-J}{\alpha M}\right)\left(\frac{1}{N}\sum_{n=1}^{N} V_{(J),n}\, \psi_n^\top T_n\right)$$
$$+ \frac{1}{\alpha M}\sum_{m=J+1}^{M}\left(\frac{1}{N}\sum_{n=1}^{N} V_{(m),n}\, \psi_n^\top T_n\right), \tag{12a}$$

$$e_{k+1} := \left(1 - \frac{M-J}{\alpha M}\right)\left(\frac{1}{N}\sum_{n=1}^{N} V_{(J),n}\, \psi_n^\top h_n\right)$$
$$+ \frac{1}{\alpha M}\sum_{m=J+1}^{M}\left(\frac{1}{N}\sum_{n=1}^{N} V_{(m),n}\, \psi_n^\top h_n\right). \tag{12b}$$

We then add the cut $E_{k+1}x + \eta \geq e_{k+1}$ to (7d), set $k \leftarrow k+1$, and resolve the master problem. The following theorem shows that the adapted L-shaped algorithm converges.

**Theorem 4.1.** *The adapted L-shaped algorithm terminates in finitely many iterations and returns an optimal solution of* (5).

Our adapted procedure extends the classical L-shaped method by exploiting the specific structure of APUB. The sorting-based computation of $\gamma(x)$ allows us to efficiently handle a large number of bootstrap samples in the second stage.

## 5. Numerical Analyses

We consider a two-stage product-mix problem with uncertain labor, adapted from the benchmark in King (1988). The goal is to choose a product mix that minimizes contracting cost and expected labor cost under uncertainty. Let $\mathcal{I}$

be the set of products and $\mathcal{J}$ the set of departments. The APUB-based two-stage model specializes to

$$\min_{x \geq 0} \sum_{i \in \mathcal{I}} c_i x_i + \mathbb{U}_{\text{APUB}}^\alpha\left[\mathbb{E}[Q(x,\xi)] \,\big|\, \widehat{\mathbb{P}}_N\right],$$

where the recourse problem is

$$Q(x,\xi) = \min_{y,z} \quad \sum_{j \in \mathcal{J}} q_j y_j$$
$$\text{s.t.} \quad w_j y_j + z_j \geq \sum_{i \in \mathcal{I}} t_{ij} x_i, \quad j \in \mathcal{J},$$
$$\sum_{j \in \mathcal{J}} z_j = h_1,$$
$$\sum_{j \in \mathcal{J}} y_j \leq h_2,$$
$$y_j \geq 0,\ z_j \geq 0, \qquad j \in \mathcal{J}.$$

In the first stage, $x_i$ denotes the contracted quantity of product $i$ with unit cost $c_i$. In the second stage, $\xi = (h, q, w)$ collects the random labor parameters: $h = (h_1, h_2)$ is the number of available permanent employees and temporary workers, $q_j$ is the cost of a temporary worker in department $j$, and $w_j$ is the efficiency of such a worker. Decisions $y_j$ and $z_j$ represent the numbers of temporary and permanent workers allocated to department $j$, and $t_{ij}$ is the labor requirement of product $i$ in department $j$.

**Data-generating process.** We model two regimes: a regular period with probability $p$ and a worst-case period with probability $1 - p$, representing low-probability, high-impact events (e.g., a pandemic). In regime $r \in \{r, w\}$ (regular / worst-case), the marginals follow uniform distributions

$$h_k \sim \mathcal{U}[\underline{h}_k^r, \overline{h}_k^r], \quad q_j \sim \mathcal{U}[\underline{q}_j^r, \overline{q}_j^r], \quad w_j \sim \mathcal{U}[\underline{w}_j^r, \overline{w}_j^r],$$

for $k \in \mathcal{K} := \{1, 2\}$ and $j \in \mathcal{J}$, and their dependence is modeled by a Gumbel copula with parameter $\lambda^r$ (regular) or $\lambda^w$ (worst-case). The exact parameter values for all instances are reported in Appendix G. We do not include a classical DRO baseline because random recourse makes such formulations computationally prohibitive in this setting.

Our baseline instance has $|\mathcal{I}| = 20$ products and $|\mathcal{J}| = 8$ departments. We have validated $M = 3000$ is sufficient for the convergence of the bootstrap sampling method in our case (see Appendix C) and use $M = 5000$ for all subsequent experiments in this study. Unless stated otherwise, we use Python 3.8 and Gurobi 11.0.3 on an i7-11700@2.50GHz machine. Throughout we denote by SAA-M the classical SAA model, which is a special case of APUB-M with $(1-\alpha) = 0$ (Remark 2.3).

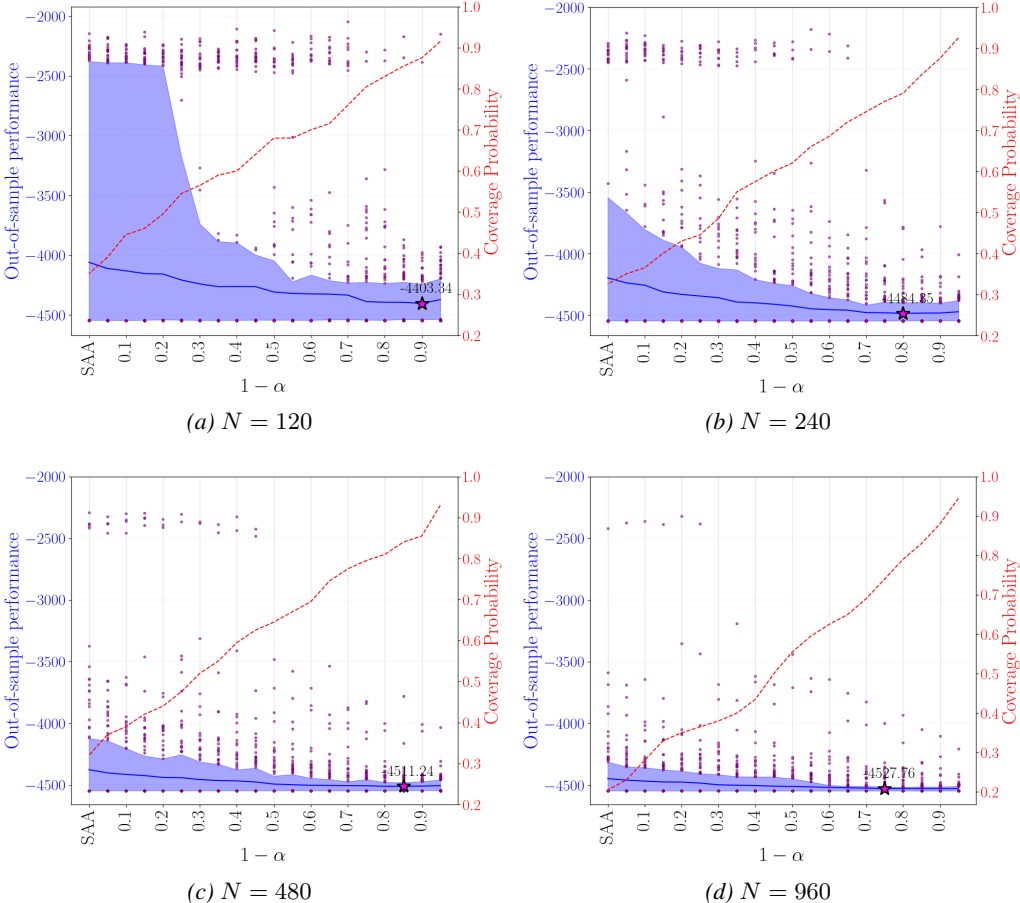

*Figure 1.* Out-of-sample performance and coverage probability of SAA-M and APUB-M for different sample sizes $N$. Solid lines: mean out-of-sample cost; shaded area: 10th–90th percentiles; dots: extremes; dashed lines: coverage probability. The star marks the minimum mean value.

## 5.1. Out-of-Sample Performance and Coverage Probability

We now compare the out-of-sample behavior of SAA-M and APUB-M. The experiment runs 200 Monte Carlo replications. In each replication we draw an i.i.d. training sample of size $N \in \{120, 240, 480, 960\}$, solve both models at various nominal levels to obtain the optimal value $\vartheta_N(\alpha)$ and first-stage solution $x_N(\alpha)$, and then estimate the true expected cost $c^\intercal x_N(\alpha) + \mathbb{E}[Q(x_N(\alpha), \xi)]$ on an independent test set of 5000 points.

Figure 1 summarizes the results. For each $N$, the solid line shows the mean out-of-sample cost, the shaded region the 10th–90th percentiles, dots indicate extreme realizations outside this band, and the dashed line reports the empirical coverage probability $\beta(\vartheta_N(\alpha), \{x_N(\alpha)\})$ in (1).

**Small samples and robustness.** For small $N$ (120 and 240), epistemic uncertainty is high and SAA-M is unstable: the mean cost is large, the percentile band is wide, and many extreme points appear. As $(1 - \alpha)$ increases, APUB-M

markedly improves both mean and worst-case performance and reduces dispersion. Notably, an appropriately chosen nominal level with $N = 120$ already outperforms SAA-M trained with $N = 240$, illustrating the robustness gains from APUB under limited data.

**Large samples and asymptotic behavior.** When $N$ grows to 480 and 960, both methods improve: mean costs decrease, bands shrink, and worst-case performance is much better than in the small-$N$ cases. The performance curves flatten as $N$ increases, and the gap between APUB-M and SAA-M narrows, in line with the asymptotic consistency of APUB-M in Theorem 3.7. This behavior also shows that APUB naturally avoids excessive conservatism when more data are available; there is no need to manually rescale the nominal level.

**Coverage probability and correctness.** The empirical coverage probability tracks the nominal level closely: for APUB-M, it increases approximately linearly with $(1 - \alpha)$ and, for large $N$, exceeds the nominal level, confirming the asymptotic correctness of Theorem 3.4. Interpreting

*Table 1.* Computational Analysis of the Baseline Problem (fixed $|\mathcal{I}| = 20$, $|\mathcal{J}| = 8$, $(1 - \alpha) = 0.9$). Mean $\pm$ std over 30 runs.

| N M | 120 | | 240 | | 480 | | 960 | |
|---|---|---|---|---|---|---|---|---|
| | Time(s) | Iteration | Time(s) | Iteration | Time(s) | Iteration | Time(s) | Iteration |
| **The standard L-shaped method solving SAA-M:** | | | | | | | | |
| | $5.7 \pm 1.0$ | $8.0 \pm 1.4$ | $12.8 \pm 1.5$ | $8.9 \pm 0.6$ | $25.3 \pm 4.5$ | $9.3 \pm 0.6$ | $47.2 \pm 6.6$ | $10.0 \pm 0.3$ |
| **Adapted L-shaped for APUB-M:** | | | | | | | | |
| 1000 | $7.5 \pm 1.2$ | $9.9 \pm 1.5$ | $17.6 \pm 2.6$ | $10.0 \pm 1.4$ | $35.8 \pm 5.3$ | $10.8 \pm 1.3$ | $73.8 \pm 7.5$ | $11.6 \pm 1.4$ |
| 3000 | $7.4 \pm 1.3$ | $9.6 \pm 1.5$ | $18.0 \pm 2.6$ | $10.2 \pm 1.3$ | $36.7 \pm 7.3$ | $11.0 \pm 1.6$ | $75.3 \pm 10.0$ | $11.8 \pm 1.3$ |
| 5000 | $7.2 \pm 1.2$ | $9.2 \pm 1.4$ | $18.1 \pm 2.4$ | $10.3 \pm 1.3$ | $36.4 \pm 5.6$ | $11.2 \pm 1.5$ | $74.1 \pm 7.8$ | $11.5 \pm 1.1$ |

*Table 2.* Computational results for varying problem dimensions ($N = 120$, $M = 5000$, $(1 - \alpha) = 0.9$). Mean $\pm$ std over 30 runs.

| $|\mathcal{I}| = 10, |\mathcal{J}| = 4$ | | $|\mathcal{I}| = 20, |\mathcal{J}| = 8$ | | $|\mathcal{I}| = 40, |\mathcal{J}| = 16$ | | $|\mathcal{I}| = 80, |\mathcal{J}| = 32$ | |
|---|---|---|---|---|---|---|---|
| Time(s) | Iteration | Time(s) | Iteration | Time(s) | Iteration | Time(s) | Iteration |
| **Standard L-shaped for SAA-M:** | | | | | | | |
| $1.3 \pm 0.1$ | $5.0 \pm 0.9$ | $5.7 \pm 1.0$ | $8.0 \pm 1.4$ | $25.3 \pm 4.5$ | $8.3 \pm 0.6$ | $58.1 \pm 7.0$ | $8.2 \pm 0.5$ |
| **Adapted L-shaped for APUB-M:** | | | | | | | |
| $2.1 \pm 0.4$ | $6.8 \pm 0.9$ | $7.2 \pm 1.2$ | $9.2 \pm 1.4$ | $36.7 \pm 7.3$ | $8.7 \pm 1.6$ | $65.3 \pm 6.6$ | $7.8 \pm 0.6$ |

$(1 - \alpha)$ as the desired confidence level therefore provides a simple, problem-agnostic guideline for tuning. In contrast, SAA-M (corresponding to $(1 - \alpha) = 0$) maintains a very low coverage probability even as $N$ grows, offering little practical value despite being technically covered by the same theorem.

Overall, APUB-M delivers significantly better out-of-sample performance and more reliable bounds than SAA-M in the data-scarce regime, while smoothly converging to similar performance as $N$ increases. Choosing the nominal level that minimizes the mean cost (the star in Figure 1) is nontrivial; in practice, standard cross-validation is a natural approach, and developing principled selection rules is an interesting direction for future work.

### 5.2. Computational Analysis

We next assess the computational performance of our adapted L-shaped Algorithm and compare it to the standard one applied to SAA-M. We consider two sets of experiments. In the first, we fix the baseline dimensions $|\mathcal{I}| = 20$, $|\mathcal{J}| = 8$ and $(1 - \alpha) = 0.9$, and vary $N \in \{120, 240, 480, 960\}$ and $M \in \{1000, \ldots, 5000\}$. In the second, we fix $N = 120$, $M = 5000$, $(1 - \alpha) = 0.9$ and vary $(|\mathcal{I}|, |\mathcal{J}|) \in \{(10, 4), (20, 8), (40, 16), (80, 32)\}$. For each configuration we use 30 randomly selected instances from those generated in Section 5.1.

Table 1 reports the mean runtime and iteration count. For both the standard L-shaped method and our adapted one, runtime scales roughly linearly with $N$, reflecting the number of second-stage subproblems, while the number of iterations grows only mildly and remains around 10. In contrast, increasing $M$ has little impact on the runtime, confirming the

effectiveness of the sorting-based evaluation of the APUB term.

Table 2 shows the effect of scaling the number of products and departments. The both methods become more expensive as $|\mathcal{I}|$ and $|\mathcal{J}|$ increase, due to larger second-stage subproblems, but the number of iterations remains small and fairly insensitive to the dimension. Across all settings, our adapted L-shaped method is only moderately slower than the standard one for SAA-M.

In summary, the adapted L-shaped method offers a favorable trade-off between robustness and computational effort. It scales well with both the number of samples and problem dimensions, making APUB-based two-stage models practical for large-scale stochastic optimization problems.

### 5.3. Comparison with Wasserstein DRO

To compare APUB-M with Wasserstein DRO (WassDRO), we fix the recourse in the second stage of the product-mix problem. Our key observations in Figure 2 are highlighted as follows:

**Statistical Interpretability vs. Geometric Tuning:** APUB-M utilizes a nominal confidence level $(1 - \alpha)$ with a rigorous frequentist interpretation. In contrast, selecting a statistically meaningful radius $\epsilon$ for WassDRO remains a significant challenge, often requiring heuristic-based calibration. As shown in Figures 2a–2c, APUB-M demonstrates asymptotic correctness, where the empirical coverage probability (dashed line) consistently converges toward the nominal confidence level $(1 - \alpha)$ as $N$ increases. This provides a transparent uncertainty knob that is missing in geometric DRO approaches.

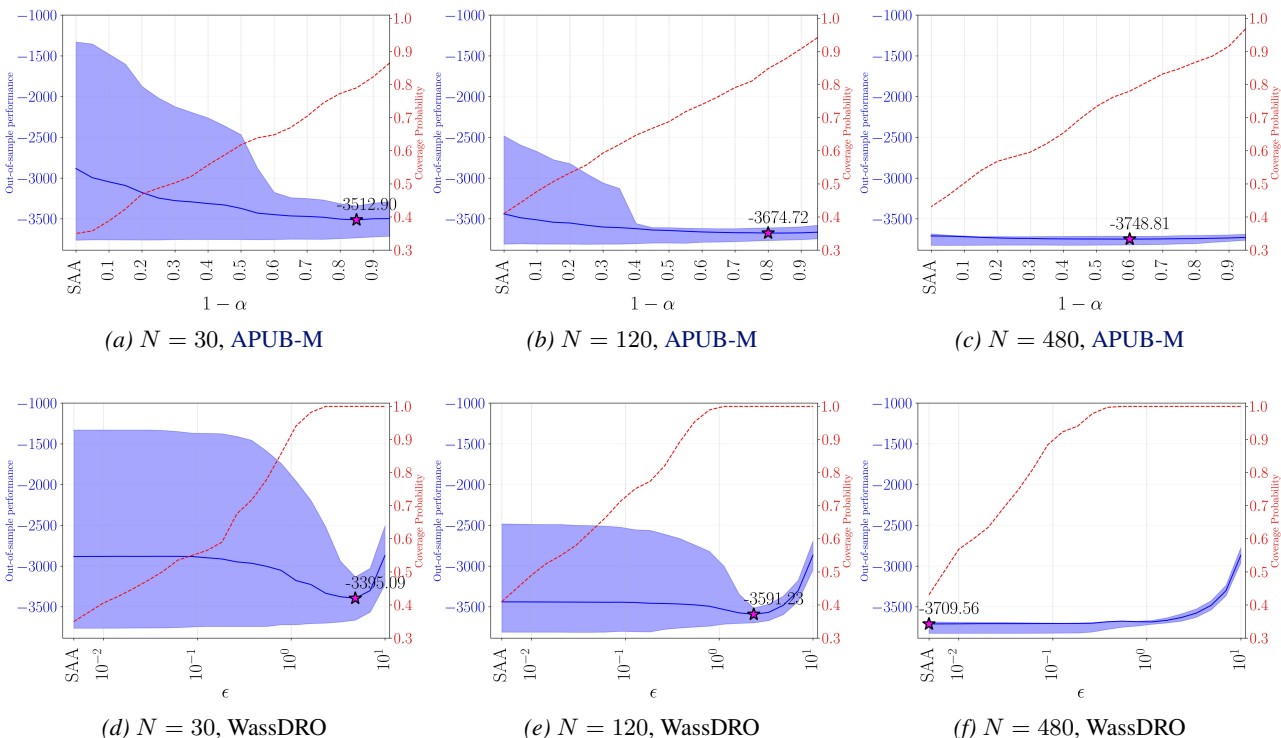

*Figure 2.* Mean out-of-sample cost (solid line), 10th–90th percentile band (shaded area), and empirical coverage probability (dashed line) for APUB-M and WassDRO. Stars mark the minimum mean cost.

**Consistent Data-Driven Convergence:** APUB-M is inherently data-driven, achieving consistency for any fixed $(1 - \alpha)$ as the bootstrap distribution naturally contracts with $N$. Conversely, WassDRO relies on a radius $\epsilon$ that typically requires manual $N$-dependent scaling to prevent vanishing consistency or extreme over-conservatism. Figures 2a–2c illustrate that APUB-M automatically modulates its robustness: as epistemic uncertainty diminishes with larger $N$, the APUB-M objective naturally converges toward the true mean without manual parameter decay.

**Controlled Conservatism and Reliability:** In low-data regimes ($N = 30$), worst-case scenarios are often underrepresented. APUB-M explicitly accounts for the epistemic uncertainty of these tail events, ensuring reliability without the structural over-conservatism seen in WassDRO. As shown in Figure 2f, an inappropriate choice of $\epsilon$ in WassDRO leads to a sharp increase in mean cost (over-conservatism). APUB-M maintains a more stable performance profile by anchoring its robustness to the statistical variance of the mean rather than an arbitrary geometric ball.

## 6. Conclusions

In this work, we introduced APUB, a novel framework for quantifying epistemic uncertainty in data-driven optimization. By leveraging a bootstrap-based CVaR representation, we established that APUB-M provides a statistically interpretable uncertainty knob that avoids the over-conservatism and manual tuning inherent in geometric DRO approaches. Our theoretical analysis proves the asymptotic correctness and consistency of the framework, while our adapted L-shaped algorithm ensures computational tractability for complex two-stage problems with random recourse. Empirical results confirm that APUB-M significantly enhances prescriptive stability and out-of-sample reliability in low-data regimes. Future research will explore the extension of the APUB framework to multi-stage stochastic programs and non-convex decision spaces.

## Acknowledgements

This work was supported by the National Science Foundation (NSF) under Award No. 2432256. The authors would like to express their sincere gratitude to Professor Jim Luedtke for his valuable observations and suggestions that helped improve the algorithm. We also thank the four anonymous reviewers for their insightful comments and constructive feedback.

## Impact Statement

This research contributes to the broader field of reliable AI by providing a mathematically grounded tool for decision-making under data scarcity. From a **societal perspective**, APUB-M enhances the resilience of critical infrastructure, such as energy grids and supply chains, where optimizer's curse failures can have significant economic or safety consequences. Regarding **accessibility**, the framework democratizes robust optimization by replacing opaque geometric parameters with transparent, frequentist confidence levels, allowing domain experts in fields like healthcare and public policy to calibrate risk without deep expertise in optimization theory. Finally, we acknowledge **ethical considerations**: while APUB-M provides safety against sampling error, it does not inherently correct for systemic biases present in historical data. We advocate for the use of this framework in conjunction with rigorous data auditing to ensure that robust prescriptions do not inadvertently perpetuate existing inequities.

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

## A. L-shaped Algorithm for the Two-Stage APUB-M

We restate the Adapted L-Shaped Algorithm in Section 4 as Algorithm 1.

---

**Algorithm 1** L-shaped algorithm for the two-stage APUB-M problem (5).

---

**Input:** Data $\{(q_n, W_n, T_n, h_n)\}_{n=1}^N$, bootstrap weights $V \in \mathbb{R}^{M \times N}$, level $\alpha \in (0, 1)$.
**Output:** Optimal first-stage solution $x^\star$.
 1: Initialize $\ell \leftarrow 0$, $k \leftarrow 0$.
 2: **while** true **do**
 3:     Solve the master problem (7) to obtain $(\hat{x}, \hat{\eta})$.
 4:     Set `infeas` $\leftarrow$ false.
 5:     **for** $n = 1, \ldots, N$ **do**
 6:         Solve the feasibility problem (10) and compute $u_n(\hat{x})$.
 7:         **if** $u_n(\hat{x}) > 0$ **then**
 8:             Extract multipliers $\phi_n$ and add the feasibility cut, $D_{\ell+1} x \geq d_{\ell+1}$, with $D_{\ell+1} = \phi_n^\top T_n$, $d_{\ell+1} = \phi_n^\top h_n$.
 9:             Update $\ell \leftarrow \ell + 1$; `infeas` $\leftarrow$ true.
10:             **break**
11:         **end if**
12:     **end for**
13:     **if** `infeas` **then**
14:         **continue** {Resolve the updated master problem.}
15:     **end if**
16:     **for** $n = 1, \ldots, N$ **do**
17:         Solve for $Q(\hat{x}, \xi_n)$ and multipliers $\psi_n$.
18:     **end for**
19:     Compute $r_m(\hat{x})$ via (9) and sort them to obtain $r_{(1)}(\hat{x}) \leq \cdots \leq r_{(M)}(\hat{x})$.
20:     Set $J \leftarrow \lceil (1 - \alpha) M \rceil$ and compute $\hat{\lambda}$ from (11).
21:     **if** $\hat{\eta} \geq \hat{\lambda}$ **then**
22:         **return** $\hat{x}$ as $x^\star$.
23:     **end if**
24:     Compute $(E_{k+1}, e_{k+1})$ using (12) and add the optimality cut, $E_{k+1} x + \eta \geq e_{k+1}$, to (7d).
25:     Update $k \leftarrow k + 1$.
26: **end while**

---

## B. Comparison between the Standard Large-Sample UCB, Efron's Percentile UCB and APUB

Let $\xi \sim \text{Gamma}(2, 1)$ and $F(\xi) = \xi$. So the population mean $\mu = 2$. We compare APUB with Efron's percentile-based upper bound and the standard large-sample upper bound given as $\hat{\mu}_N + z_\alpha \hat{\sigma}_N / \sqrt{N}$, where $z_\alpha$ denotes $z$ critical value. In order to estimate the probability density functions (pdf) of three upper bounds, we performed a Monte Carlo simulation with $\alpha = 0.05$ while allowing the sample sizes, $N$, to vary from 80 to 10,000.

We make two observations:

- **APUB exhibits asymptotic correctness as an upper confidence bound.** As the sample size increases, the coverage probability of APUB remains above the nominal level, rather than converging exactly to it. By contrast, Efron's bound and the standard upper bound become closer to the nominal level. This highlights that APUB is asymptotically correct, but not asymptotically exact.
- **APUB is asymptotically consistent.** As the sample size increases, their sampling distributions become more concentrated around the true mean. In particular, APUB becomes increasingly tight and converges to the correct limit, consistent with the asymptotic consistency established in the main paper.

## C. Convergence of Bootstrap Sampling Approximation.

We now assess the convergence of the bootstrap sampling method applied to APUB-M. Our evaluation encompasses 10 independent simulations, each producing $N = 120$ sample data points. Throughout these tests, we maintain a consistent

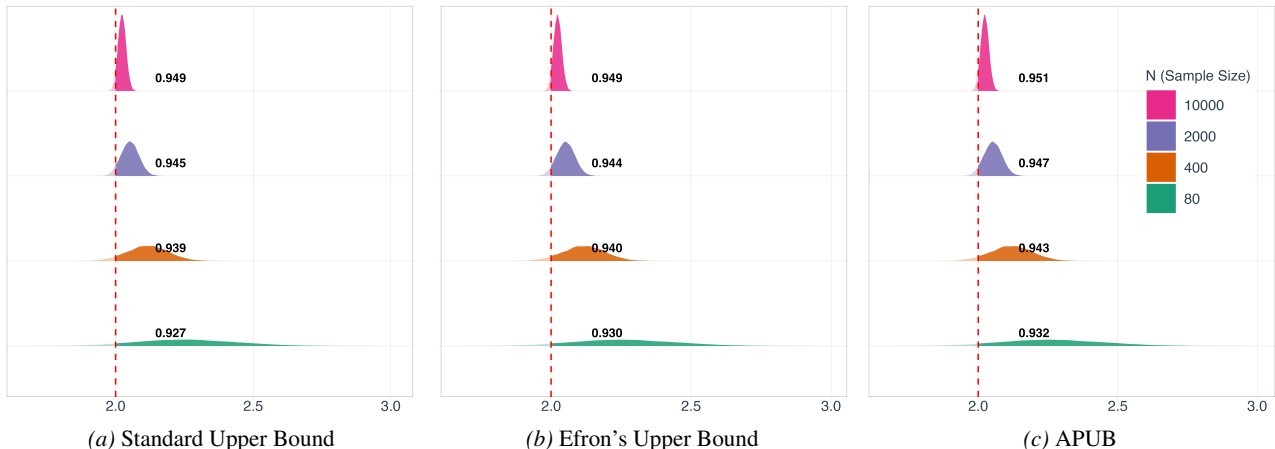

*(a)* Standard Upper Bound     *(b)* Efron's Upper Bound     *(c)* APUB

*Figure 3.* The comparison between APUB, Efron's upper bound, and the standard large-sample upper bound.

nominal level of $(1 - \alpha) = 0.9$. Figure 4 illustrates the relationship between the number $M$ of bootstrap samples and the optimal values of the bootstrap approximation, with $M$ reaching up to 8000. The data shows a clear stabilization trend: as $M$ increases, the variability in the optimal values visibly decreases. The convergence of the approximation becomes evident for $M \geq 3000$, where the fluctuation in the optimal values lessens significantly. This consistency supports our decision to use $M = 5000$ for all subsequent experiments in this section.

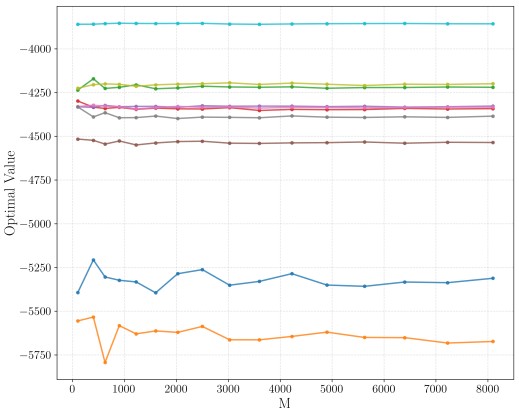

*Figure 4.* Convergence of the bootstrap sampling approximation.

## D. Sensitivity Analysis of APUB-M on Multi-Product Newsvendor Problem

Our another experiment evaluates the APUB-M framework using a 10-product newsvendor problem based on the formulation by Hanasusanto et al. (2015) (detailed parameters are provided in Appendix G.4). This benchmark is particularly relevant as Mohajerin Esfahani & Kuhn (2018) have noted that Wasserstein-based DRO may encounter performance limitations in settings like the newsvendor problem, where the random loss function possesses a Lipschitz modulus with respect to the random scenarios that is independent of the decision variables. To assess the robustness of APUB-M, we consider two distinct demand scenarios:

- **Case I:** Demand follows a standard Gaussian-mixture distribution, representing a multimodal but well-behaved environment.
- **Case II:** Demand is subjected to additional biased noise, creating higher variance and a more complex underlying true distribution to simulate a high-ambiguity environment.

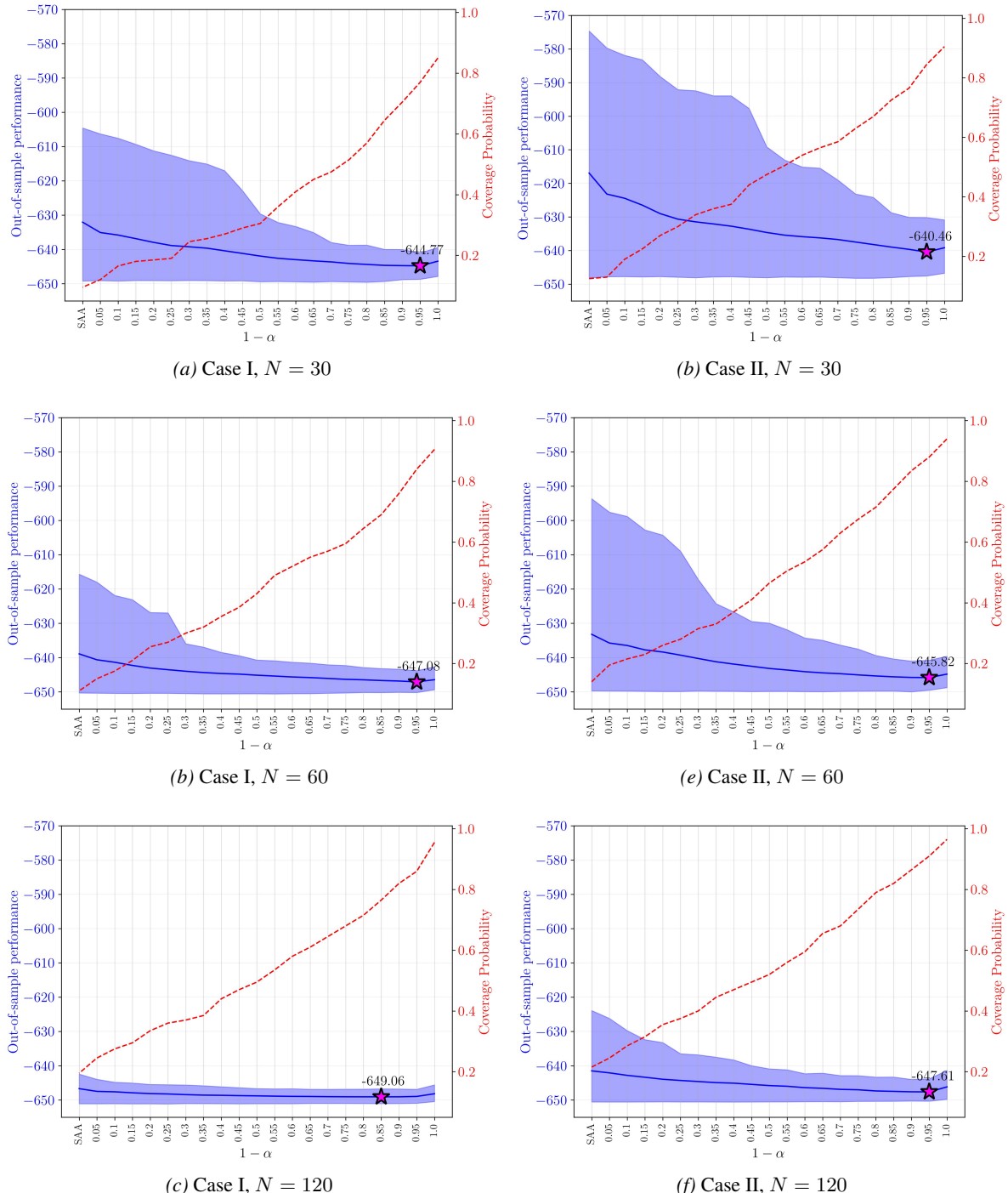

*Figure 5.* Mean out-of-sample cost (solid line), 10th–90th percentile band (shaded area), and empirical coverage probability (dashed line) for APUB-M under two ambiguity levels. Stars mark the minimum mean cost.

## D.1. Out-of-Sample Behavior

Figure 5 summarizes the out-of-sample behavior of APUB-M across two experimental settings. The primary trend remains consistent across all panels: APUB-M consistently outperforms SAA-M in both mean cost and performance variability. These gains are most pronounced when data are scarce ($N = 30$) or when the underlying distribution exhibits high ambiguity. Specifically, Case II represents a more challenging environment than Case I. At $N = 30$, the performance band in Case II is significantly wider, and the nominal confidence level that minimizes the mean cost is notably higher. This suggests that heightened ambiguity necessitates a more conservative APUB objective to maintain reliability. As the sample size increases from $N = 30$ to $N = 120$, the performance gap between the two cases narrows, demonstrating that APUB-M effectively adapts to decreasing epistemic uncertainty while continuing to leverage the information in larger datasets.

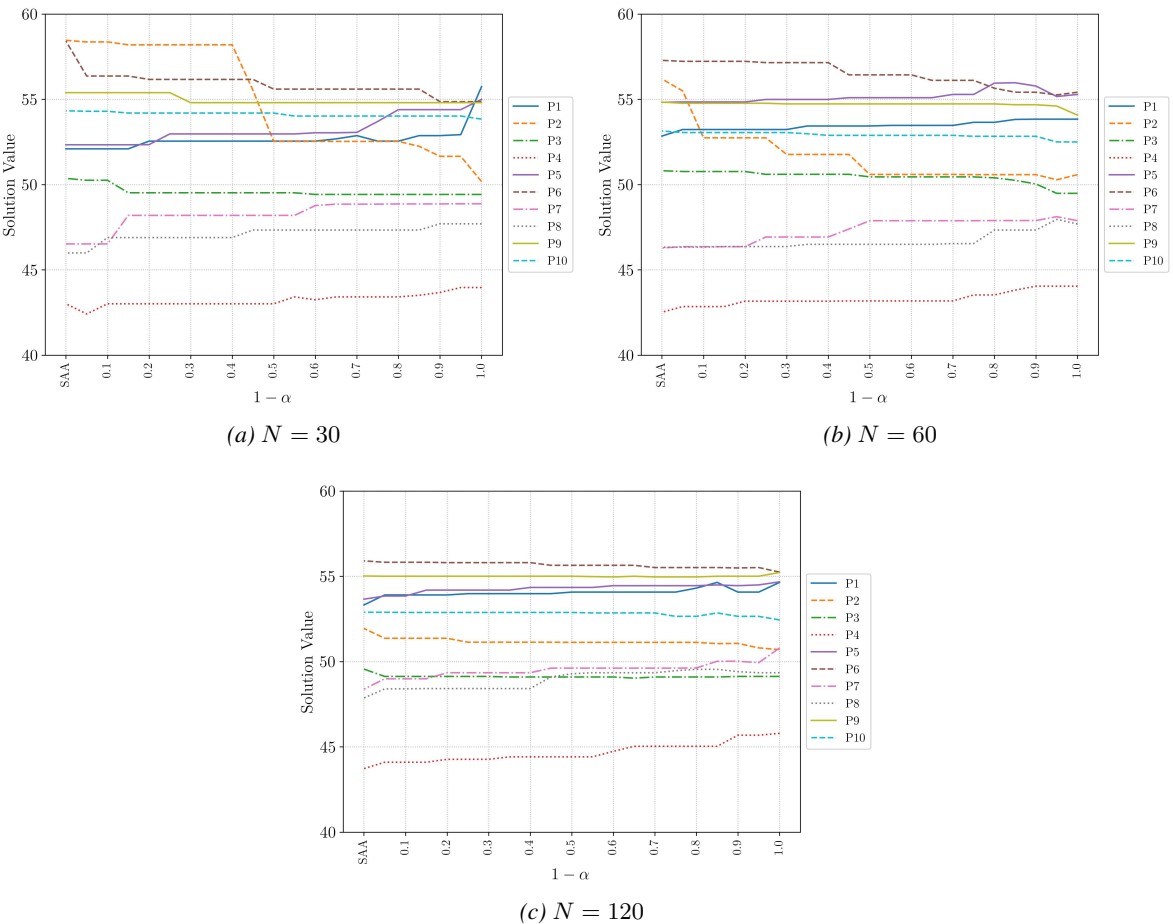

*(a) $N = 30$*

*(b) $N = 60$*

*(c) $N = 120$*

*Figure 6.* Optimal order quantities for the 10 products.

## D.2. Stability of Optimal Solutions

Figure 6 visualizes the resulting order quantities for Case I. When data are scarce ($N = 30$), the optimal solution is highly sensitive to the nominal level ($1 - \alpha$), particularly for specific products (e.g., P2). As $N$ increases, these sensitivity curves flatten, and the APUB-M solutions smoothly converge toward the SAA-M estimates. This behavior mirrors the performance trends and reinforces the theoretical consistency of the framework. A critical advantage of APUB-M is the prescriptive stability of its solutions across varying sample sizes. To quantify this, we measure the change in the recommended order vector from $N = 30$ to $N = 120$ using the $\ell_2$-norm. For SAA-M, the shift is 7.89. In contrast, APUB-M yields significantly more stable decisions, with a shift of only 3.99 at $(1 - \alpha) = 0.5$ and 3.36 at $(1 - \alpha) = 0.95$. These results indicate that APUB-M provides materially more reliable and consistent prescriptions in data-poor environments, reducing the nervousness of the optimization results as more data are collected.

# E. Proofs of Theorems and Propositions

### E.1. Proof of Proposition 2.4

Efron's percentile-based upper bound is asymptotically accurate to the first-order, i.e.,

$$\Pr\left(\mu \le U_{\text{Efron}}^{\alpha}\left[\mu \mid \widehat{\mathbb{P}}_N\right]\right) = (1 - \alpha) + O(N^{-1/2}).$$

Its proof refers to Section 4.2 in (Shao & Tu, 2012). By Definition 2.1, we know that $\mathbb{U}_{\text{APUB}}^{\alpha}[\mu|\widehat{\mathbb{P}}_N] \ge \mathbb{U}_{\text{Efron}}^{\alpha}[\mu|\widehat{\mathbb{P}}_N]$. □

### E.2. Proof of Theorem 2.5

To prove Theorem 2.5, we need the following lemma about the bootstrap law of large numbers.

**Lemma E.1.** *Let* $(\zeta_1, \dots, \zeta_N) \sim \widehat{\mathbb{P}}_N$. *Then, as* $N \to \infty$, $\frac{1}{N} \sum_{n=1}^{N} F(\zeta_n) \to \mu$ *w.p.1.*

*Proof.* According to Theorem 2 in (Athreya, 1983) (Theorem F.2), if $\liminf_{M,N\to\infty} MN^{-\phi} > 0$ for some $\phi > 0$, and $\mathbb{E}|F(\xi) - \mu|^{\theta} < \infty$ for some $\theta \ge 1$ such that $\theta\phi > 1$, we have that, as $M, N \to \infty$, $\frac{1}{M} \sum_{m=1}^{M} F(\zeta_m) \to \mu$ w.p.1, where $(\zeta_1, \dots, \zeta_M) \sim \widehat{\mathbb{P}}_N$. Choose $\phi = 1$, $\theta = 2$, and $M = N$. It ensures $\liminf_{M,N\to\infty} MN^{-\phi} = 1 > 0$. The condition $\mathbb{E}|Z(\xi) - \mu|^{\theta} < \infty$ is satisfied due to finite variance. This completes the proof. □

**Proof of Theorem 2.5.** Let $(\bar{\xi}_1, \bar{\xi}_2, \dots)$ be a realization of the sample path and let $\bar{\mathbb{P}}_N$ be the empirical distribution associated with the first $N$ sample points. Let $(\zeta_1(\bar{\mathbb{P}}_N), \dots, \zeta_N(\bar{\mathbb{P}}_N)) \sim \bar{\mathbb{P}}_N$ be a bootstrap sample. Define the event

$$\mathfrak{S} := \left\{ (\bar{\xi}_1, \bar{\xi}_2, \dots) \; : \; \lim_{N\to\infty} \frac{1}{N} \sum_{n=1}^{N} F(\bar{\xi}_n) = \mu, \quad \lim_{N\to\infty} \frac{1}{N} \sum_{n=1}^{N} F(\zeta_n(\bar{\mathbb{P}}_N)) = \mu \text{ w.p.1 (for } \zeta) \right\}. \tag{A-1}$$

By the strong law of large numbers and Lemma E.1, we have $\Pr\{(\xi_1, \xi_2, \dots) \in \mathfrak{S}\} = 1$.

Fix an arbitrary $(\bar{\xi}_1, \bar{\xi}_2, \dots) \in \mathfrak{S}$ and its associated empirical measures $(\bar{\mathbb{P}}_1, \bar{\mathbb{P}}_2, \dots)$. Then $(\bar{\xi}_1, \bar{\xi}_2, \dots)$ and $\bar{\mathbb{P}}_N$ are deterministic, while $(\zeta_1(\bar{\mathbb{P}}_N), \dots, \zeta_N(\bar{\mathbb{P}}_N))$ remains random. For clarity, define the bootstrap sample mean

$$\widehat{\mu}_N(\bar{\mathbb{P}}_N) := \frac{1}{N} \sum_{n=1}^{N} F(\zeta_n(\bar{\mathbb{P}}_N)).$$

By Proposition 2.2, we have

$$\mathbb{U}_{\text{APUB}}^{\alpha}[\mu \mid \bar{\mathbb{P}}_N] = \text{CVaR}_{\alpha}(\widehat{\mu}_N(\bar{\mathbb{P}}_N) \mid \bar{\mathbb{P}}_N),$$

where is the CVaR of $\widehat{\mu}_N(\bar{\mathbb{P}}_N)$ conditional on $\bar{\mathbb{P}}_N$. Hence it suffices to show

$$\lim_{N\to\infty} \text{CVaR}_{\alpha}(\widehat{\mu}_N(\bar{\mathbb{P}}_N) \mid \bar{\mathbb{P}}_N) = \mu. \tag{A-2}$$

**Step 1** $L^1$-**convergence of** $\widehat{\mu}_N(\bar{\mathbb{P}}_N)$. Conditional on $\bar{\mathbb{P}}_N$, the bootstrap draws $\zeta_1(\bar{\mathbb{P}}_N), \dots, \zeta_N(\bar{\mathbb{P}}_N)$ are i.i.d. with common distribution $\bar{\mathbb{P}}_N$. Hence, by linearity of conditional expectation,

$$\mathbb{E}[\widehat{\mu}_N(\bar{\mathbb{P}}_N) \mid \bar{\mathbb{P}}_N] = \mathbb{E}\left[\frac{1}{N} \sum_{n=1}^{N} F(\zeta_n(\bar{\mathbb{P}}_N)) \Big| \bar{\mathbb{P}}_N\right] = \frac{1}{N} \sum_{n=1}^{N} \mathbb{E}[F(\zeta_n(\bar{\mathbb{P}}_N)) \mid \bar{\mathbb{P}}_N].$$

Because $\zeta_n(\bar{\mathbb{P}}_N) \sim \bar{\mathbb{P}}_N$, we have

$$\mathbb{E}[F(\zeta_n(\bar{\mathbb{P}}_N)) \mid \bar{\mathbb{P}}_N] = \int F(\xi) \, \bar{\mathbb{P}}_N(d\xi) = \frac{1}{N} \sum_{i=1}^{N} F(\bar{\xi}_i),$$

and therefore

$$\mathbb{E}[\widehat{\mu}_N(\bar{\mathbb{P}}_N) \mid \bar{\mathbb{P}}_N] = \frac{1}{N} \sum_{i=1}^{N} F(\bar{\xi}_i).$$

Next we derive the conditional variance, where variance operator is denoted by $\mathbb{V}(\cdot)$. Conditional on $\bar{\mathbb{P}}_N$, the bootstrap draws $\zeta_1(\bar{\mathbb{P}}_N), \ldots, \zeta_N(\bar{\mathbb{P}}_N)$ are i.i.d. from $\bar{\mathbb{P}}_N$, hence

$$\mathbb{V}\big(\widehat{\mu}_N(\bar{\mathbb{P}}_N) \mid \bar{\mathbb{P}}_N\big) = \mathbb{V}\left( \frac{1}{N} \sum_{n=1}^{N} F\big(\zeta_n(\bar{\mathbb{P}}_N)\big) \,\bigg|\, \bar{\mathbb{P}}_N \right) = \frac{1}{N} \mathbb{V}\big(F\big(\zeta_1(\bar{\mathbb{P}}_N)\big) \mid \bar{\mathbb{P}}_N\big).$$

Moreover, since $\zeta_1(\bar{\mathbb{P}}_N) \sim \bar{\mathbb{P}}_N$,

$$\mathbb{V}\big(F\big(\zeta_1(\bar{\mathbb{P}}_N)\big) \mid \bar{\mathbb{P}}_N\big) = \int F(\xi)^2 \, \bar{\mathbb{P}}_N(d\xi) - \left( \int F(\xi) \, \bar{\mathbb{P}}_N(d\xi) \right)^2$$

$$= \frac{1}{N} \sum_{i=1}^{N} F(\bar{\xi}_i)^2 - \left( \frac{1}{N} \sum_{i=1}^{N} F(\bar{\xi}_i) \right)^2 =: \hat{\sigma}_N^2(\bar{\mathbb{P}}_N).$$

Therefore,

$$\mathbb{V}\big(\widehat{\mu}_N(\bar{\mathbb{P}}_N) \mid \bar{\mathbb{P}}_N\big) = \frac{\hat{\sigma}_N^2(\bar{\mathbb{P}}_N)}{N}.$$

Now use the conditional mean-square decomposition:

$$\mathbb{E}\big[(\widehat{\mu}_N(\bar{\mathbb{P}}_N) - \mu)^2 \mid \bar{\mathbb{P}}_N\big] = \mathbb{V}\big(\widehat{\mu}_N(\bar{\mathbb{P}}_N) \mid \bar{\mathbb{P}}_N\big) + \left( \mathbb{E}[\widehat{\mu}_N(\bar{\mathbb{P}}_N) \mid \bar{\mathbb{P}}_N] - \mu \right)^2$$

$$= \frac{\hat{\sigma}_N^2(\bar{\mathbb{P}}_N)}{N} + \left( \frac{1}{N} \sum_{i=1}^{N} F(\bar{\xi}_i) - \mu \right)^2.$$

Since $(\bar{\xi}_1, \bar{\xi}_2, \ldots) \in \mathfrak{S}$, we have $\frac{1}{N} \sum_{i=1}^{N} F(\bar{\xi}_i) \to \mu$. Under the standing assumption $\mathbb{E}[F(\xi_1)^2] < \infty$, the strong law applied to $F(\xi_1)^2$ implies $\hat{\sigma}_N^2(\bar{\mathbb{P}}_N) \to \sigma^2 < \infty$, and hence $\hat{\sigma}_N^2(\bar{\mathbb{P}}_N)/N \to 0$. Consequently,

$$\mathbb{E}\big[(\widehat{\mu}_N(\bar{\mathbb{P}}_N) - \mu)^2 \mid \bar{\mathbb{P}}_N\big] \to 0.$$

Finally, by Cauchy–Schwarz,

$$\mathbb{E}\Big[\big|\widehat{\mu}_N(\bar{\mathbb{P}}_N) - \mu\big| \,\Big|\, \bar{\mathbb{P}}_N\Big] \leq \sqrt{\mathbb{E}\Big[(\widehat{\mu}_N(\bar{\mathbb{P}}_N) - \mu)^2 \,\Big|\, \bar{\mathbb{P}}_N\Big]} \to 0,$$

which implies $\mathbb{E}\big[\big|\widehat{\mu}_N(\bar{\mathbb{P}}_N) - \mu\big|\big] \to 0$ (with respect to the bootstrap randomness, conditional on $\bar{\mathbb{P}}_N$).

**Step 2: A Lipschitz bound for** $\mathrm{CVaR}_\alpha$. Recall the Rockafellar–Uryasev representation: for any integrable random variable $Z$,

$$\mathrm{CVaR}_\alpha(Z) = \inf_{t \in \mathbb{R}} \left\{ t + \frac{1}{\alpha} \mathbb{E}[(Z - t)_+] \right\}.$$

Using $(a)_+ - (b)_+ \leq (a - b)_+$, for any integrable $X, Y$ we have

$$\mathrm{CVaR}_\alpha(X) - \mathrm{CVaR}_\alpha(Y) \leq \left( t + \frac{1}{\alpha} \mathbb{E}[(X - t)_+] \right) - \left( t + \frac{1}{\alpha} \mathbb{E}[(Y - t)_+] \right)$$

$$= \frac{1}{\alpha} \mathbb{E}[(X - t)_+ - (Y - t)_+] \leq \frac{1}{\alpha} \mathbb{E}[(X - Y)_+] \leq \frac{1}{\alpha} \mathbb{E}|X - Y|,$$

where $t \in \mathbb{R}$ is arbitrary. Swapping $X$ and $Y$ yields

$$|\mathrm{CVaR}_\alpha(X) - \mathrm{CVaR}_\alpha(Y)| \leq \frac{1}{\alpha} \mathbb{E}|X - Y|.$$

**Step 3: Conclude** (A-2). Apply the Lipschitz bound with $X = \widehat{\mu}_N(\bar{\mathbb{P}}_N)$ and $Y = \mu$ (a constant). Since $\mathrm{CVaR}_\alpha(\mu) = \mu$, we obtain

$$\big|\mathrm{CVaR}_\alpha\big(\widehat{\mu}_N(\bar{\mathbb{P}}_N) \mid \bar{\mathbb{P}}_N\big) - \mu\big| \leq \frac{1}{\alpha} \mathbb{E}\Big[\big|\widehat{\mu}_N(\bar{\mathbb{P}}_N) - \mu\big| \,\Big|\, \bar{\mathbb{P}}_N\Big] \to 0$$

by Step 1. This proves (A-2). Since the chosen sample path $(\bar{\xi}_1, \bar{\xi}_2, \ldots) \in \mathfrak{S}$ was arbitrary and $\Pr\{(\xi_1, \xi_2, \ldots) \in \mathfrak{S}\} = 1$, the claim follows. □

## E.3. Proof of Theorem 3.4

For the convenience of reading, we restate some notations here. Recall $(\xi_1, \ldots, \xi_N)$ is an i.i.d. random sample from $(\Omega, \mathcal{F}, \mathbb{P})$. Consider a function $F : \mathcal{X} \times \Xi \to \mathbb{R}$, where $\Xi$ is the support of $\xi_1$ and $\mathcal{X}$ is a decision region. We define the population mean $\mu(x) = \mathbb{E}[F(x, \xi)]$ and variance $\sigma^2(x) = \mathbb{E}[(F(x, \xi) - \mu(x))^2]$. Let $\widehat{\mathbb{P}}_N$ be the empirical distribution associated with the random sample, and $(\zeta_1, \ldots, \zeta_N)$ is a bootstrap sample generated from $\widehat{\mathbb{P}}_N$. Accordingly, denote $\widehat{\mu}_N(x) = \mathbb{E}[F(x, \zeta)|\widehat{\mathbb{P}}_N]$, $\widehat{\sigma}_N^2(x) = \mathbb{E}\left[(F(x, \zeta) - \widehat{\mu}_N(x))^2 \,\middle|\, \widehat{\mathbb{P}}_N\right]$, and the standardized sample mean

$$S_N(x) := \frac{\sqrt{N}\,(\widehat{\mu}_N(x) - \mu(x))}{\sigma(x)}.$$

Denote the bootstrap counterparts of $\widehat{\mu}_N(x)$ and $S_N(x)$ by

$$\widehat{\mu}_N^*(x) := \frac{1}{N} \sum_{n=1}^{N} F(x, \zeta_n) \quad \text{and} \quad S_N^*(x) := \frac{\sqrt{N}\,(\widehat{\mu}_N^*(x) - \widehat{\mu}_N(x))}{\widehat{\sigma}_N(x)}.$$

We now outline our proof as follows. We know by Proposition E.9 that, the objective function of APUB-M, $\mathbb{U}_{\text{APUB}}^{\alpha}[\mu(x)|\widehat{\mathbb{P}}_N]$, is continuous. Also, Assumption 3.1 states that the set $\widehat{\mathcal{S}}_N^{\alpha}$ of the optimal solutions of APUB-M is contained in the compact set $\mathcal{K}$ for a sufficiently large $N$ w.p.1. This implies that $\widehat{\mathcal{S}}_N^{\alpha}$ is compact and then there exists $\tilde{x}_N \in \widehat{\mathcal{S}}_N^{\alpha}$ such that $\mu(\tilde{x}_N) = \max_{x \in \widehat{\mathcal{S}}_N^{\alpha}} \mu(x)$. Subsequently, we

$$\beta(\widehat{\vartheta}_N^{\alpha}, \widehat{\mathcal{S}}_N^{\alpha}) = \Pr\left(\mathbb{U}_{\text{APUB}}^{\alpha}[\mu(\tilde{x}_N)|\widehat{\mathbb{P}}_N] \geq \mu(\tilde{x}_N)\right) \geq \Pr\left(\mathbb{U}_{\text{Efron}}^{\alpha}[\mu(\tilde{x}_N)|\widehat{\mathbb{P}}_N] \geq \mu(\tilde{x}_N)\right),$$

which means that, to obtain $\lim_{N \to \infty} \beta(\widehat{\vartheta}_N^{\alpha}, \widehat{\mathcal{S}}_N^{\alpha}) \geq (1 - \alpha)$, it suffices to prove the following uniform convergence as

$$\lim_{N \to \infty} \sup_{x \in \mathcal{K}} \left| \Pr\left(\mathbb{U}_{\text{Efron}}^{\alpha}[\mu(x)|\widehat{\mathbb{P}}_N] \geq \mu(x)\right) - (1 - \alpha) \right| = 0. \tag{A-3}$$

In this section we describe the proof of equation (A-3) in five steps as follows:

  i) Section E.3.1 shows the first three moments of $F(x, \xi)$ are finite.

  ii) Section E.3.2 proves that as $N$ increases to $\infty$, the cdf of $S_N(x)$ uniformly (in $x \in \mathcal{K}$) approximates to the standard normal distribution function.

  iii) Section E.3.3 proves a bootstrap version of the result of step (iii).

  iv) Section E.3.4 links $\mathbb{U}_{\text{Efron}}^{\alpha}[\mu(x)|\widehat{\mathbb{P}}_N]$ to $S_N(x)$, and utilizes the results from the step (ii) and (iii) to prove the bootstrap percentile uniformly converges to normal percentile.

  v) Section E.3.5 rigorously proves (A-3).

### E.3.1. Finite Moments and Domination Property.

The following lemma shows the first three moments of $F(x, \xi)$ are finite.

**Lemma E.2.** *Under the assumptions of Theorem 3.4, for $k \leq 3$, there exist $K_k$ such that $\sup_{x \in \mathcal{K}} \mathbb{E}\left[|F(x, \xi)|^k\right] \leq K_k$.*

*Proof.* We first prove the boundedness for $\mathbb{E}\left[|F(x, \xi)|^k\right]$ over $x \in \mathcal{K}$. The Lipschitz continuity of $F(x, \xi)$ leads to $|F(x, \xi)| \leq |F(x_0, \xi)| + |F(x, \xi) - F(x_0, \xi)| \leq |F(x_0, \xi)| + L(\xi)\|x - x_0\| \leq |F(x_0, \xi)| + D \cdot L(\xi)$, where $D := \sup_{x \in \mathcal{K}} \|x - x_0\|$ is the diameter of $\mathcal{K}$. Raising both sides to the power $k$ and applying the inequality $(a + b)^k \leq 2^{k-1}(a^k + b^k)$ for $a, b \geq 0$, we obtain $|F(x, \xi)|^k \leq 2^{k-1}\left(|F(x_0, \xi)|^k + D^k L(\xi)^k\right)$. Taking expectations on both sides, $\mathbb{E}\left[|F(x, \xi)|^k\right] \leq 2^{k-1}\left(\mathbb{E}\left[|F(x_0, \xi)|^k\right] + D^k \mathbb{E}\left[L(\xi)^k\right]\right)$. The assumptions of Theorem 3.4 requires that $\mathbb{E}\left[|F(x_0, \xi)|^k\right] < \infty$ and $\mathbb{E}\left[L(\xi)^k\right] < \infty$ for $k \leq 3$. By letting $K_k := 2^{k-1}\left(\mathbb{E}\left[|F(x_0, \xi)|^k\right] + D^k \mathbb{E}\left[L(\xi)^k\right]\right)$, we complete the proof. $\square$

**Lemma E.3.** *Under the assumptions of Theorem 3.4, there exists an integrable function $G(x, \xi)$ with $\mathbb{E}[G(x, \xi)] < \infty$ for each $x \in \mathcal{K}$ such that $|F(x, \xi)| \leq G(x, \xi)$.*

*Proof.* By Assumption (ii) of Theorem 3.4,

$$|F(x, \xi)| \leq |F(x, \xi) - F(x_0, \xi)| + |F(x_0, \xi)| < |F(x_0, \xi)| + L(\xi)\|x - x_0\|.$$

Define $G(x, \xi) := |F(x_0, \xi)| + L(\xi)\|x - x_0\|$. Then, we have $\mathbb{E}[G(x, \xi)] = \mathbb{E}|F(x_0, \xi)| + \mathbb{E}[L(\xi)]\|x - x_0\|$. Assumption (i) and (iii) of Theorem 3.4 implies $\mathbb{E}|F(x_0, \xi)| < \infty$ and $\mathbb{E}[L(\xi)] < \infty$. Thus, $\mathbb{E}[G(x, \xi)] < \infty$ for all $x \in \mathcal{K}$. □

### E.3.2. UNIFORM BERRY-ESSEEN INEQUALITY.

The following theorem states the Berry-Esseen bound for $S_N(x)$, which is fundamental in our proof.

**Theorem E.4** (Berry-Esseen Inequality (Berry, 1941)). *For a fixed $x \in \mathcal{K}$, if $\mathbb{E}|F(x, \xi)|^3 < \infty$, then*

$$\sup_{z \in \mathbb{R}} \left| \Pr(S_N(x) \leq z) - \Phi(z) \right| \leq C \frac{\mathbb{E}|F(x, \xi)|^3}{\sigma^3(x)\sqrt{N}}$$

*where $C$ is a constant independent of $x$.*

**Lemma E.5.** *Under the assumptions of Theorem 3.4, we have $\Pr(S_N(x) \leq z) = \Phi(z) + o(1)$ uniformly in $x \in \mathcal{K}$ and $z \in \mathbb{R}$.*

*Proof.* It follows by the assumptions of Theorem 3.4 that there exist a constant $\sigma_{\min}$ such that $\sigma(x) \geq \sigma_{\min}$ for all $x \in \mathcal{K}$. In addition, by Lemma E.2, $\sup_{x \in \mathcal{K}} \mathbb{E}|F(x, \xi)|^3 \leq K_3$. Then, by Theorem E.4, we have

$$\sup_{x \in \mathcal{K}} \sup_{z \in \mathbb{R}} |\Pr(S_N(x) \leq z) - \Phi(z)| \leq C \frac{\sup_{x \in \mathcal{K}} \mathbb{E}|F(x, \xi)|^3}{\inf_{x \in \mathcal{K}} \sigma^3(x)\sqrt{N}} \leq C \frac{K_3}{\sigma_{min}^3 \sqrt{N}} = o(1).$$

□

### E.3.3. UNIFORM BERRY-ESSEEN INEQUALITY FOR PERCENTILE BOOTSTRAP SAMPLING.

We now show the uniform asymptotic behavior of the moments of $\widehat{\mathbb{P}}_N$ in the following lemma.

**Lemma E.6.** *Under the assumptions of Theorem 3.4, the following two conditions hold as $N \to \infty$ w.p.1.,*

(i)    $\sup_{x \in \mathcal{K}} \left| \mathbb{E}\left[ |F(x, \zeta)|^k \,\middle|\, \widehat{\mathbb{P}}_N \right] - \mathbb{E}\left[ |F(x, \xi)|^k \right] \right| \to 0$ *for $k \leq 3$.*

(ii)    $\sup_{x \in \mathcal{X}} |\widehat{\sigma}_N(x) - \sigma(x)| \to 0$.

*Proof.* We now prove (i). Assumption 3.2 states that $F(x, \xi)$ is continuous in $x \in \mathcal{K}$ for any $\xi$. In addition, Lemma E.3 implies $F(x, \xi)$ dominated by an integrable function for each $x \in \mathcal{K}$. According to Theorem 7.53 in (Shapiro et al., 2021) (Theorem F.5), we know that, as $N \to \infty$,

$$\sup_{x \in \mathcal{K}} \left| \mathbb{E}\left[ |F(x, \zeta)|^k \,\middle|\, \widehat{\mathbb{P}}_N \right] - \mathbb{E}\left[ |F(x, \xi)|^k \right] \right| \to 0, \quad \text{w.p.1.} \tag{A-4}$$

We next prove (ii). Since

$$\begin{aligned}
\widehat{\sigma}_N^2(x) &= \mathbb{E}\left[ (F(x, \zeta) - \widehat{\mu}_N(x))^2 \,\middle|\, \widehat{\mathbb{P}}_N \right] \\
&= \mathbb{E}\left[ ((F(x, \zeta) - \mu(x)) - (\widehat{\mu}_N(x) - \mu(x)))^2 \,\middle|\, \widehat{\mathbb{P}}_N \right] \\
&= \mathbb{E}\left[ (F(x, \zeta) - \mu(x))^2 \,\middle|\, \widehat{\mathbb{P}}_N \right] - (\widehat{\mu}_N(x) - \mu(x))^2,
\end{aligned}$$

we obtain

$$\sup_{x\in\mathcal{K}} \left|\widehat{\sigma}_N^2(x) - \sigma^2(x)\right| \leq \sup_{x\in\mathcal{K}} \left|\mathbb{E}\left[(F(x,\zeta) - \mu(x))^2 \,\Big|\, \widehat{\mathbb{P}}_N\right] - \sigma^2(x)\right| + \sup_{x\in\mathcal{K}} \left(\widehat{\mu}_N(x) - \mu(x)\right)^2.$$

Equation (A-4) has shown that the two terms at the right-hand-side of the above inequality converge to 0 as $N \to \infty$ w.p.1, which implies (ii) holds. $\square$

The following lemma shows the Berry-Esseen Inequality for the percentile bootstrap approach.

**Lemma E.7.** *Under the assumptions of Theorem 3.4,* $\Pr\left(S_N^*(x) \leq z \,\Big|\, \widehat{\mathbb{P}}_N\right) = \Phi(z) + o(1)$ *uniformly in* $x \in \mathcal{K}$ *and* $z \in \mathbb{R}$ *w.p.1.*

*Proof.* By the assumptions of Theorem 3.4, we know there exist $\sigma_{min}$ such that $\sigma(x) > \sigma_{min} > 0$. In addition, by Lemma E.2, $\sup_{x\in\mathcal{K}} \mathbb{E}\left[|F(x,\xi)|^3\right] \leq K_3$. By Lemma E.6, the following two inequalities hold w.p.1,

$$\lim_{N\to\infty} \sup_{x\in\mathcal{K}} \mathbb{E}\left[|F(x,\zeta)|^3 \,\Big|\, \widehat{\mathbb{P}}_N\right]$$

$$\leq \lim_{N\to\infty} \sup_{x\in\mathcal{K}} \left|\mathbb{E}\left[|F(x,\zeta)|^3 \,\Big|\, \widehat{\mathbb{P}}_N\right] - \mathbb{E}\,|F(x,\xi)|^3\right| + \sup_{x\in\mathcal{K}} \mathbb{E}\,|F(x,\xi)|^3 \leq K_3,$$

$$\lim_{N\to\infty} \inf_{x\in\mathcal{K}} |\widehat{\sigma}_N(x)| \geq \inf_{x\in\mathcal{K}} |\sigma(x)| - \lim_{N\to\infty} \sup_{x\in\mathcal{K}} |\widehat{\sigma}_N(x) - \sigma(x)| \geq \sigma_{min}.$$

Then, by Lemma E.4, we have

$$\lim_{N\to\infty} \sup_{x\in\mathcal{K}} \sup_{z\in\mathbb{R}} \left|\Pr\left(S_N^*(x) \leq z \,\Big|\, \widehat{\mathbb{P}}_N\right) - \Phi(z)\right| \leq \lim_{N\to\infty} C \frac{\sup_{x\in\mathcal{K}} \mathbb{E}\left[|F(x,\xi)|^3 \,\Big|\, \widehat{\mathbb{P}}_N\right]}{\inf_{x\in\mathcal{K}} \widehat{\sigma}_N^3(x)\sqrt{N}}$$

$$\leq \lim_{N\to\infty} C \frac{K_3}{\sigma_{min}^3 \sqrt{N}} = 0.$$

$\square$

### E.3.4. UNIFORM CONVERGENCE ON BOOTSTRAP PERCENTILE.

Define

$$z_N^\alpha(x) := \inf_z \left\{z \in \mathbb{R} : \Pr\left(S_N^*(x) \leq z \,\Big|\, \widehat{\mathbb{P}}_N\right) \geq 1 - \alpha\right\}.$$

Recall

$$\mathbb{U}_{\text{Efron}}^\alpha[\mu(x)|\widehat{\mathbb{P}}_N] = \inf\left\{z \in \mathbb{R} : \Pr\left(\widehat{\mu}_N^*(x) \leq z \,\Big|\, \widehat{\mathbb{P}}_N\right) \geq 1 - \alpha\right\}.$$

We derive

$$\mathbb{U}_{\text{Efron}}^\alpha[\mu(x)|\widehat{\mathbb{P}}_N] = \widehat{\mu}_N(x) + \frac{\widehat{\sigma}_N(x)}{\sqrt{N}} z_N^\alpha(x).$$

Letting

$$T_N^\alpha(x) := -\frac{\widehat{\sigma}_N(x)}{\sigma(x)} \cdot z_N^\alpha(x),$$

we further express the coverage probability as

$$\Pr\left(\mathbb{U}_{\text{Efron}}^\alpha[\mu(x)|\widehat{\mathbb{P}}_N] \geq \mu(x)\right) = \Pr\left(\widehat{\mu}_N(x) + \frac{\widehat{\sigma}_N(x)}{\sqrt{N}} \cdot z_N^\alpha(x) \geq \mu(x)\right)$$

$$= \Pr\left(\frac{\sqrt{N}(\widehat{\mu}_N(x) - \mu(x))}{\sigma(x)} \geq -\frac{\widehat{\sigma}_N(x)}{\sigma(x)} \cdot z_N^\alpha(x)\right) = \Pr\left(S_N(x) \geq T_N^\alpha(x)\right). \tag{A-5}$$

**Lemma E.8.** $\lim_{N\to\infty} \sup_{x\in\mathcal{K}} |T_N^\alpha(x) - \Phi^{-1}(\alpha)| = 0, \quad$ *w.p.1.*

*Proof.* Consider a sample path $\boldsymbol{\xi} = (\xi_1, \xi_2, \dots)$ and denote its sample space by $\Xi^\infty$. Define

$$\Upsilon := \left\{ \boldsymbol{\xi} \in \Xi^\infty : \lim_{N \to \infty} \sup_{x \in \mathcal{K}} |T_N^\alpha(x) - \Phi^{-1}(\alpha)| = 0 \right\},$$

$$\Upsilon_0 := \left\{ \boldsymbol{\xi} \in \Xi^\infty : \lim_{N \to \infty} \sup_{x \in \mathcal{K}} \left| \frac{\widehat{\sigma}_N(x)}{\sigma(x)} - 1 \right| = 0 \right\},$$

$$\Upsilon_1 := \left\{ \boldsymbol{\xi} \in \Xi^\infty : \lim_{N \to \infty} \sup_{x \in \mathcal{K}} \left| (-z_N^\alpha(x)) - \Phi^{-1}(\alpha) \right| = 0 \right\},$$

and

$$\Upsilon_2 := \left\{ \boldsymbol{\xi} \in \Xi^\infty : \lim_{N \to \infty} \sup_{x \in \mathcal{K}} \sup_{z \in \mathbb{R}} \left| \Pr\left( S_N^*(x) \le z \mid \widehat{\mathbb{P}}_N \right) - \Phi(z) \right| = 0 \right\}.$$

Our goal is to prove $\Pr(\boldsymbol{\xi} \in \Upsilon) = 1$. We now give our proof in the following three steps.

**i) Prove that** $(\Upsilon_0 \cap \Upsilon_1) \subseteq \Upsilon$**.** Consider $\boldsymbol{\xi} \in \Upsilon_0 \cap \Upsilon_1$. Observe that

$$\left| T_N^\alpha(x) - \Phi^{-1}(\alpha) \right| = \left| \left( \frac{\widehat{\sigma}_N(x)}{\sigma(x)} - 1 \right) (-z_N^\alpha(x)) + \left( -z_N^\alpha(x) - \Phi^{-1}(\alpha) \right) \right|$$

$$\le \left| \frac{\widehat{\sigma}_N(x)}{\sigma(x)} - 1 \right| \left| -z_N^\alpha(x) \right| + \left| -z_N^\alpha(x) - \Phi^{-1}(\alpha) \right|.$$

By definition of $\Upsilon_1$, we know $-z_N^\alpha(x) \to \Phi^{-1}(\alpha)$ uniformly in $x \in \mathcal{K}$. There exists $N_1$ such that for all $N \ge N_1$,

$$\sup_{x \in \mathcal{K}} \left| -z_N^\alpha(x) \right| \le \left| \Phi^{-1}(\alpha) \right| + 1.$$

Similarly, by definition of $\Upsilon_0$ and $\Upsilon_1$, for any $\varepsilon > 0$, there exists $N_2$ such that for all $N \ge N_2$,

$$\sup_{x \in \mathcal{K}} \left| \frac{\widehat{\sigma}_N(x)}{\sigma(x)} - 1 \right| < \varepsilon, \quad \text{and} \quad \sup_{x \in \mathcal{K}} \left| -z_N^\alpha(x) - \Phi^{-1}(\alpha) \right| < \varepsilon.$$

Therefore, for $N \ge N_0 := \max\{N_1, N_2\}$, we have

$$\left| T_N^\alpha(x) - \Phi^{-1}(\alpha) \right| \le \varepsilon \left( \left| \Phi^{-1}(\alpha) \right| + 1 \right) + \varepsilon.$$

Since $\varepsilon > 0$ is arbitrary, it follows that

$$\lim_{N \to \infty} \sup_{x \in \mathcal{K}} \left| T_N^\alpha(x) - \Phi^{-1}(\alpha) \right| = 0.$$

Thus, $\boldsymbol{\xi} \in \Upsilon$, and therefore $(\Upsilon_0 \cap \Upsilon_1) \subseteq \Upsilon$, which completes the proof for Step i).

**ii) Prove that** $\Upsilon_2 \subseteq \Upsilon_1$**.** Consider $\boldsymbol{\xi} \in \Upsilon_2$ and a given $\delta > 0$. Since $\Phi(z)$ is continuous and strictly increasing, its inverse $\Phi^{-1}(\cdot)$ is also continuous and strictly increasing on its domain $(0, 1)$. Let

$$\varepsilon(\delta) := \min \left\{ \Phi\left( \Phi^{-1}(1 - \alpha) + \delta \right) - (1 - \alpha), \ (1 - \alpha) - \Phi\left( \Phi^{-1}(1 - \alpha) - \delta \right) \right\}.$$

By definition of $\Upsilon_2$, there exists $N_0$ such that for all $N \ge N_3$, we have

$$\sup_{x \in \mathcal{K}} \sup_{z \in \mathbb{R}} \left| \Pr\left( S_N^*(x) \le z \mid \widehat{\mathbb{P}}_N \right) - \Phi(z) \right| < \varepsilon(\delta).$$

In other words, for all $N \ge N_3$, we have:

$$\left| \Pr\left( S_N^*(x) \le z \mid \widehat{\mathbb{P}}_N \right) - \Phi(z) \right| < \varepsilon(\delta), \quad \text{for all } x \in \mathcal{K}, \ z \in \mathbb{R}.$$

Thus, at $z = \Phi^{-1}(1 - \alpha) - \delta$:

$$\Pr\left(S_N^*(x) \leq \Phi^{-1}(1 - \alpha) - \delta \,\Big|\, \widehat{\mathbb{P}}_N\right) < \Phi\left(\Phi^{-1}(1 - \alpha) - \delta\right) + \varepsilon(\delta)$$

$$\leq (1 - \alpha) - \varepsilon(\delta) + \varepsilon(\delta) = 1 - \alpha.$$

Similarly, at $z = \Phi^{-1}(1 - \alpha) + \delta$:

$$\Pr\left(S_N^*(x) \leq \Phi^{-1}(1 - \alpha) + \delta \,\Big|\, \widehat{\mathbb{P}}_N\right) > \Phi\left(\Phi^{-1}(1 - \alpha) + \delta\right) - \varepsilon(\delta)$$

$$\geq (1 - \alpha) + \varepsilon(\delta) - \varepsilon(\delta) = 1 - \alpha.$$

Therefore, it follows by the definition of $z_N^\alpha(x)$ that, for all $N \geq N_3$ and $x \in \mathcal{K}$, $z_N^\alpha(x) \in \left[\Phi^{-1}(1 - \alpha) - \delta, \ \Phi^{-1}(1 - \alpha) + \delta\right]$. Thus,

$$\sup_{x \in \mathcal{K}} \left|z_N^\alpha(x) - \Phi^{-1}(1 - \alpha)\right| \leq \delta.$$

Note that $\Phi^{-1}(\alpha) = -\Phi^{-1}(1 - \alpha)$. When choosing an arbitrarily small $\delta$, we obtain

$$\lim_{N \to \infty} \sup_{x \in \mathcal{K}} \left|(-z_N^\alpha(x)) - \Phi^{-1}(\alpha)\right| = 0,$$

which completes the proof for Step ii).

**iii) Prove that** $\Pr(\boldsymbol{\xi} \in \Upsilon) = 1$**.** By in Lemma E.6, it is clear to see $\Pr(\boldsymbol{\xi} \in \Upsilon_0) = 1$. In addition, by Lemma E.7, $\Pr(\xi \in \Upsilon_2) = 1$. By previous two steps, we establish

$$(\Upsilon_0 \cap \Upsilon_2) \subseteq (\Upsilon_0 \cap \Upsilon_1) \subseteq \Upsilon.$$

Therefore, $\Pr(\boldsymbol{\xi} \in \Upsilon) = 1$. It completes the proof of Lemma E.8. $\qquad\square$

### E.3.5. PROOF OF EQUATION (A-3).

Recall from (A-5) that for every $x \in \mathcal{K}$,

$$\Pr\left(\mathbb{U}_{\mathrm{Efron}}^\alpha[\mu(x) \mid \widehat{\mathbb{P}}_N] \geq \mu(x)\right) = \Pr(S_N(x) \geq T_N^\alpha(x)).$$

Hence it suffices to prove

$$\lim_{N \to \infty} \sup_{x \in K} |\Pr(S_N(x) \geq T_N^\alpha(x)) - (1 - \alpha)| = 0. \tag{A-6}$$

Let $c := \Phi^{-1}(\alpha)$ so that $1 - \Phi(c) = 1 - \alpha$. Fix an arbitrary $\varepsilon > 0$ and define the event

$$B_{N,\varepsilon}(x) := \{|T_N^\alpha(x) - c| \leq \varepsilon\}, \qquad x \in \mathcal{K}.$$

On $B_{N,\varepsilon}(x)$ we have $c - \varepsilon \leq T_N^\alpha(x) \leq c + \varepsilon$.

**Step 1: Two-sided sandwich bounds for each fixed $x$.** First, on the event $B_{N,\varepsilon}(x)$,

$$\{S_N(x) \geq T_N^\alpha(x)\} \cap B_{N,\varepsilon}(x) \subseteq \{S_N(x) \geq c - \varepsilon\}.$$

Therefore,

$$\Pr(S_N(x) \geq T_N^\alpha(x)) = \Pr(\{S_N(x) \geq T_N^\alpha(x)\} \cap B_{N,\varepsilon}(x)) + \Pr(\{S_N(x) \geq T_N^\alpha(x)\} \cap B_{N,\varepsilon}(x)^c)$$

$$\leq \Pr(S_N(x) \geq c - \varepsilon) + \Pr(B_{N,\varepsilon}(x)^c)$$

$$= \Pr(S_N(x) \geq c - \varepsilon) + \Pr(|T_N^\alpha(x) - c| > \varepsilon). \tag{A-7}$$

Similarly, since $T_N^\alpha(x) \leq c + \varepsilon$ on $B_{N,\varepsilon}(x)$,

$$\{S_N(x) \geq c + \varepsilon\} \cap B_{N,\varepsilon}(x) \subseteq \{S_N(x) \geq T_N^\alpha(x)\},$$

and thus

$$\Pr(S_N(x) \geq T_N^\alpha(x)) \geq \Pr(S_N(x) \geq c + \varepsilon) - \Pr(B_{N,\varepsilon}(x)^c)$$

$$= \Pr(S_N(x) \geq c + \varepsilon) - \Pr(|T_N^\alpha(x) - c| > \varepsilon). \tag{A-8}$$

**Step 2: Take $\sup_{x \in \mathcal{K}}$ and reduce the threshold term to a uniform event.** Note that for every $x \in \mathcal{K}$,

$$\{|T_N^\alpha(x) - c| > \varepsilon\} \subseteq \left\{\sup_{u \in \mathcal{K}} |T_N^\alpha(u) - c| > \varepsilon\right\},$$

hence

$$\sup_{x \in \mathcal{K}} \Pr(|T_N^\alpha(x) - c| > \varepsilon) \leq \Pr\left(\sup_{u \in \mathcal{K}} |T_N^\alpha(u) - c| > \varepsilon\right). \tag{A-9}$$

Combining (A-7)–(A-8) with (A-9) yields

$$\sup_{u \in \mathcal{K}} \Pr(S_N(x) \geq T_N^\alpha(x)) \leq \sup_{u \in \mathcal{K}} \Pr(S_N(x) \geq c - \varepsilon) + \Pr\left(\sup_{u \in \mathcal{K}} |T_N^\alpha(u) - c| > \varepsilon\right), \tag{A-10}$$

$$\inf_{u \in \mathcal{K}} \Pr(S_N(x) \geq T_N^\alpha(x)) \geq \inf_{u \in \mathcal{K}} \Pr(S_N(x) \geq c + \varepsilon) - \Pr\left(\sup_{u \in \mathcal{K}} |T_N^\alpha(u) - c| > \varepsilon\right). \tag{A-11}$$

**Step 3: Apply Lemma E.8 and Lemma E.5.** By Lemma E.8,

$$\sup_{u \in \mathcal{K}} |T_N^\alpha(u) - c| \to 0 \quad \text{w.p.1},$$

which implies for every fixed $\varepsilon > 0$,

$$\Pr\left(\sup_{u \in \mathcal{K}} |T_N^\alpha(u) - c| > \varepsilon\right) \to 0. \tag{A-12}$$

By Lemma E.5, for any fixed $t \in \mathbb{R}$,

$$\sup_{x \in \mathcal{K}} |\Pr(S_N(x) \geq t) - (1 - \Phi(t))| \to 0.$$

In particular, for $t = c - \varepsilon$,

$$\sup_{x \in \mathcal{K}} \Pr(S_N(x) \geq c - \varepsilon) \to 1 - \Phi(c - \varepsilon), \tag{A-13}$$

and for $t = c + \varepsilon$,

$$\inf_{x \in \mathcal{K}} \Pr(S_N(x) \geq c + \varepsilon) \to 1 - \Phi(c + \varepsilon). \tag{A-14}$$

**Step 4: Conclude the uniform convergence.** Taking $\limsup_{N \to \infty}$ in (A-10) and $\liminf_{N \to \infty}$ in (A-11), and using (A-12)–(A-14), we obtain

$$\limsup_{N \to \infty} \sup_{x \in \mathcal{K}} \Pr(S_N(x) \geq T_N^\alpha(x)) \leq 1 - \Phi(c - \varepsilon),$$

$$\liminf_{N \to \infty} \inf_{x \in \mathcal{K}} \Pr(S_N(x) \geq T_N^\alpha(x)) \geq 1 - \Phi(c + \varepsilon).$$

Therefore, for every $\varepsilon > 0$,

$$\limsup_{N \to \infty} \sup_{x \in \mathcal{K}} \left| \Pr(S_N(x) \geq T_N^\alpha(x)) - (1 - \Phi(c)) \right| \leq \max\left\{\Phi(c) - \Phi(c - \varepsilon), \ \Phi(c + \varepsilon) - \Phi(c)\right\}.$$

Letting $\varepsilon \downarrow 0$ and using the continuity of $\Phi$ yields

$$\lim_{N \to \infty} \sup_{x \in \mathcal{K}} |\Pr(S_N(x) \geq T_N^\alpha(x)) - (1 - \Phi(c))| = 0.$$

Since $1 - \Phi(c) = 1 - \alpha$, this proves (A-6), and hence Equation (A-3). $\qquad\square$

### E.4. Proof of Theorem 3.7

To prove Theorem 3.7, we first prove the following two lemmas: the first lemma show the continuity of $\mu(x)$ and $\mathbb{U}_{\text{APUB}}^{\alpha}[\mu(x)|\widehat{\mathbb{P}}_N]$; the second lemma shows the uniform consistency of $\mathbb{U}_{\text{APUB}}^{\alpha}[\mu(x)|\widehat{\mathbb{P}}_N]$ on $\mathcal{K}$.

**Lemma E.9.** *Suppose Assumption 3.2 holds. Then $\mu(x)$ is continuous on $\mathcal{N}$, and $\mathbb{U}_{APUB}^{\alpha}[\mu(x)|\widehat{\mathbb{P}}_N]$ is a continuous convex function on $\mathcal{N}$ w.p.1.*

*Proof.* Since $F(x, \xi)$ is convex on $\mathcal{N}$, $\mu(x)$ is also convex on $\mathcal{N}$. Hence, $\mu(x)$ is continuous. Under Assumption 3.2, $F(\cdot, \xi)$ is continuous and convex on $\mathcal{N}$. It is easy to see that, by Proposition 2.2, $\mathbb{U}_{\text{APUB}}^{\alpha}[\mu(x)|\widehat{\mathbb{P}}_N]$ is continuous convex on $\mathcal{N}$. $\square$

**Lemma E.10.** *Suppose Assumption 3.1 and 3.2 holds. Then, we have as $N \to \infty$,*

$$\sup_{x \in \mathcal{K}} \left| \mathbb{U}_{APUB}^{\alpha}[\mu(x)|\widehat{\mathbb{P}}_N] - \mu(x) \right| \to 0, \ \text{w.p.1.}$$

*Proof.* Note that the open convex set $\mathcal{N} \subseteq \mathbb{R}^{d_x}$. We first construct a countable dense subset of $\mathcal{N}$ as $\mathcal{D} := \mathbb{Q}^{d_x} \cap \mathcal{N}$, where $\mathbb{Q}^{d_x}$ represents the set of $d_x$-dimensional rational numbers. Choose a sample path $(\xi_1, \xi_2, \dots)$ and hence $\widehat{\mathbb{P}}_N$ is the empirical distribution associated to the first $N$ sample points. For $x \in \mathcal{D}$, we denote an event as

$$\Upsilon_x := \left\{ (\xi_1, \xi_2, \dots) \ : \ \lim_{N \to \infty} \mathbb{U}_{\text{APUB}}^{\alpha}[\mu(x)|\widehat{\mathbb{P}}_N] = \mu(x) \right\}.$$

Since $\mu(x) < \infty$ and $\sigma(x) < \infty$ under Assumption 3.2, it follows by Theorem 2.5 that $\Pr\left((\xi_1, \xi_2, \dots) \in \Upsilon_x\right) = 1$, which implies that $\Pr\left((\xi_1, \xi_2, \dots) \in \bigcap_{x \in \mathcal{D}} \Upsilon_x\right) = 1$. In other words, $\mathbb{U}_{\text{APUB}}^{\alpha}[\mu(x)|\widehat{\mathbb{P}}_N]$ converges pointwisely to $\mu(x)$ on $\mathcal{D}$ w.p.1. Furthermore, by Proposition E.9 and Theorem 10.8 in (Rockafellar, 2015) (Theorem F.4), we can conclude that $\mathbb{U}_{\text{APUB}}^{\alpha}[\mu(x)|\widehat{\mathbb{P}}_N]$ converges uniformly a certain continuous function $\nu$ on $\mathcal{K}$ w.p.1. Since $\nu(x)$ and $\mu(x)$ coincidence on a dense subset of $\mathcal{K}$ and they are both continuous on $\mathcal{K}$, we know that $\nu(x) = \mu(x)$ for all $x \in \mathcal{K}$. This completes the proof. $\square$

### Proof of Theorem 3.7

**i) Proof of the consistency of $\widehat{\vartheta}_N^{\alpha}$.** Choose $x^* \in \mathcal{S}$ and $\widehat{x}_N \in \widehat{\mathcal{S}}_N$. Then,

$$\mathbb{U}_{\text{APUB}}^{\alpha}[\mu(\widehat{x}_N)|\widehat{\mathbb{P}}_N] \le \mathbb{U}_{\text{APUB}}^{\alpha}[\mu(x^*)|\widehat{\mathbb{P}}_N] \quad \text{and} \quad \mu(x^*) \le \mu(\widehat{x}_N).$$

Thus, we have

$$
\begin{aligned}
|\widehat{\vartheta}_N^{\alpha} - \vartheta^*| &= \left| \mathbb{U}_{\text{APUB}}^{\alpha}[\mu(\widehat{x}_N)|\widehat{\mathbb{P}}_N] - \mu(x^*) \right| \\
&= \max\left\{ \mathbb{U}_{\text{APUB}}^{\alpha}[\mu(\widehat{x}_N)|\widehat{\mathbb{P}}_N] - \mu(x^*), \ \mu(x^*) - \mathbb{U}_{\text{APUB}}^{\alpha}[\mu(\widehat{x}_N)|\widehat{\mathbb{P}}_N] \right\}, \\
&\le \max\left\{ \mathbb{U}_{\text{APUB}}^{\alpha}[\mu(x^*)|\widehat{\mathbb{P}}_N] - \mu(x^*), \ \mu(\widehat{x}_N) - \mathbb{U}_{\text{APUB}}^{\alpha}[\mu(\widehat{x}_N)|\widehat{\mathbb{P}}_N] \right\} \\
&\le \sup_{x \in \mathcal{K}} \left| \mathbb{U}_{\text{APUB}}^{\alpha}[\mu(x)|\widehat{\mathbb{P}}_N] - \mu(x) \right|,
\end{aligned}
$$

which converges to 0 w.p.1 by Theorem E.10. This completes the proof.

**ii) Proof of the consistency of $\widehat{\mathcal{S}}_N^{\alpha}$.** Let $\mathcal{O}$ as a collection of sample paths along which $\widehat{\mathcal{S}}_N^{\alpha} \subseteq \mathcal{K}$ for a sufficiently large $N$ and $\widehat{\vartheta}_N^{\alpha} \to \vartheta^*$. By the above proof and Assumption 3.1, we have $\Pr\left((\xi_1, \xi_2, \dots) \in \mathcal{O}\right) = 1$. We now choose $(\xi_1, \xi_2, \dots) \in \mathcal{O}$. Thus $\widehat{\mathcal{S}}_N^{\alpha}$ is the optimal solution set of APUB-M using the first $N$ sample points.

Suppose by contradiction that $\mathbb{D}(\widehat{\mathcal{S}}_N^{\alpha}, \mathcal{S}) \not\to 0$ along the sample path $(\xi_1, \xi_2, \dots)$. Then, there exists $\varepsilon > 0$ such that for all $M \in \mathbb{N}$, there exists some $N > M$ for which $\mathbb{D}(\widehat{\mathcal{S}}_N^{\alpha}, \mathcal{S}) > \varepsilon$. Specifically, there exists $\widehat{x}_N \in \widehat{\mathcal{S}}_N^{\alpha}$ such that $\inf_{y \in \mathcal{S}} \|\widehat{x}_N, y\| > \varepsilon$. Because of the compactness of $\mathcal{K}$, we can find a subsequence $\widehat{x}_{N_k} \in \widehat{\mathcal{S}}_{N_k}^{\alpha}$ such that $\widehat{x}_{N_k} \subseteq \mathcal{K}$ for all $k \in \mathbb{N}$, and $\lim_{k \to \infty} \widehat{x}_{N_k} = \widehat{x} \in \mathcal{K}, \inf_{y \in \mathcal{S}} \|\widehat{x}_N, y\| > \varepsilon$, for all $k$. It follows that $\widehat{x} \notin \mathcal{S}$ and hence $\mu(\widehat{x}) > \vartheta^*$. On the other hand, we have

$$\left| \mathbb{U}_{\text{APUB}}^{\alpha}[\mu(\widehat{x}_{N_k})|\widehat{\mathbb{P}}_N] - \mu(\widehat{x}) \right| \le \left| \mathbb{U}_{\text{APUB}}^{\alpha}[\mu(\widehat{x}_{N_k})|\widehat{\mathbb{P}}_N] - \mu(\widehat{x}_{N_k}) \right| + \left| \mu(\widehat{x}_{N_k}) - \mu(\widehat{x}) \right|.$$

On the right hand of the above inequality, the first term converges to zero by Theorem E.10, and the second term converges to zero because of the continuity of $\mu(x)$. Thus,

$$\lim_{k\to\infty} \mathbb{U}^{\alpha}_{\mathrm{APUB}}[\mu(\widehat{x}_{N_k})|\widehat{\mathbb{P}}_N] = \mu(\widehat{x}).$$

The definition of $\mathcal{O}$ ensures that $\mathbb{U}^{\alpha}_{\mathrm{APUB}}[\mu(\widehat{x}_{N_k})|\widehat{\mathbb{P}}_N] = \widehat{\vartheta}^{\alpha}_N \to \vartheta^*$. It implies that $\mu(\widehat{x}) = \vartheta^*$. This is contradictory to the assertion that $\mathbb{D}(\widehat{\mathcal{S}}^{\alpha}_N, \mathcal{S}) \not\to 0$.

### E.5. Proof of Theorem 4.1

**Key observation:**  Consider a combination without replacement which firstly selects an element in $\{1, \ldots, M\}$ and next pick out other $M - J$ different elements from the remains. We depict this combination as a set $\{(i_1), (i_2, \ldots, i_{M-J+1})\}$, where each $i_j$ is unique and belongs to $\{1, \ldots, M\}$. In this way, we can form $\binom{M}{1}\binom{M-1}{M-J}$ different sets, denoted as $\Gamma_{\ell}$ for $\ell = 1, \ldots, \binom{M}{1}\binom{M-1}{M-J}$. For $\Gamma_{\ell} = \{(i_1), (i_2, \ldots, i_{M-J+1})\}$, we write $\overline{\Gamma}_{\ell} = i_1$ and $\underline{\Gamma}_{\ell} = \{i_2, \ldots, i_{M-J+1}\}$. The weighted average for $\Gamma_{\ell}$ is represented as

$$\tau_{\ell}(x) := \left(1 - \frac{M-J}{\alpha M}\right) r_{\overline{\Gamma}_{\ell}}(x) + \frac{1}{\alpha M}\sum_{j\in\underline{\Gamma}_{\ell}} r_j(x). \tag{A-15}$$

Then, as the weighted average of the extreme losses, $\gamma(x)$ equals to the maximum of $\tau_{\ell}(x)$ for all combinations $\ell \in \left\{1, \ldots, \binom{M}{1}\binom{M-1}{M-J}\right\}$, i.e.,

$$\gamma(x) = \max_{\ell\in\left\{1,\ldots,\binom{M}{1}\binom{M-1}{M-J}\right\}} \tau_{\ell}(x). \tag{A-16}$$

**Reformulation of problem** (A-17)   Define the two sets

$$\mathcal{K}_1 := \{x : Ax = b, x \geq 0\}$$

and

$$\mathcal{K}_2 := \{x : \text{there exists some } y \geq 0, \text{ s.t.} W_n y = h_n - T_n x, \text{ for all } n = 1, \ldots, N\}.$$

Their intersection, $\mathcal{K} := \mathcal{K}_1 \cap \mathcal{K}_2$, consists of all the first-stage solutions of problem (5), with which the second-stage is feasible. Notice that Algorithm 1 allows $\mathcal{K}_2 = \varnothing$. By the relations described in (8) and (A-16), we can reformulate problem (5) as

$$\min\{c^{\mathsf{T}}x + \eta : \quad x \in \mathcal{K}, \text{ and } \eta \geq \tau_{\ell}(x) \text{ for all } \ell \in \mathfrak{L}\}, \tag{A-17}$$

where $\mathfrak{L} := \left\{1, \ldots, \binom{M}{1}\binom{M-1}{M-J}\right\}$ and $\tau_{\ell}(\cdot)$ is defined in (A-15).

Let $\mathcal{E}_n$ be the set of all extreme points of $\{\pi_n : \pi_n^{\mathsf{T}}W_n \leq q_n\}$ for $n = 1, \ldots, N$, and let their Cartesian product be $\mathcal{E} := \prod_{n=1}^N \mathcal{E}_n$. It follows by the property of strong duality that $Q(x, \xi_n) = \max_{\pi_n\in\mathcal{E}_n}\{\pi_n^{\mathsf{T}}(h_n - T_n x)\}$, and then we rewrite equation (A-15) as

$$\begin{aligned}
&\tau_{\ell}(x)\\
&= \left(1 - \frac{M-J}{\alpha M}\right)\frac{1}{N}\sum_{n=1}^N V_{\overline{\Gamma}_{\ell},n}Q(x, \xi_n) + \frac{1}{\alpha M N}\sum_{j\in\underline{\Gamma}_{\ell}}\frac{1}{N}\sum_{n=1}^N V_{j,n}Q(x, \xi_n)\\
&= \max_{(\pi_1,\ldots,\pi_N)\in\mathcal{E}}\left\{\left(1 - \frac{M-J}{\alpha M}\right)\frac{1}{N}\sum_{n=1}^N V_{\overline{\Gamma}_{\ell},n}\pi_n^{\mathsf{T}}(h_n - T_n x)\right.\\
&\qquad\qquad \left. + \frac{1}{\alpha M N}\sum_{j\in\underline{\Gamma}_{\ell}}\sum_{n=1}^N V_{j,n}\pi_n^{\mathsf{T}}(h_n - T_n x)\right\}\\
&= \max_{\pi\in\mathcal{E}}\lambda_{\ell,\pi}(x),
\end{aligned}$$

where

$$\lambda_{\ell,\pi}(x) := \left(1 - \frac{M-J}{\alpha M}\right) \frac{1}{N} \sum_{n=1}^{N} V_{\overline{\Gamma}_\ell,n} \pi_n^{\mathsf{T}}(h_n - T_n x) + \frac{1}{\alpha MN} \sum_{j \in \underline{\Gamma}_\ell} \sum_{n=1}^{N} V_{j,n} \pi_n^{\mathsf{T}}(h_n - T_n x),$$

for $\pi = (\pi_1, \ldots, \pi_N) \in \mathcal{E}$. We can then reformulate problem (A-17) as

$$\min_{x,\eta} \{c^{\mathsf{T}}x + \eta : \quad x \in \mathcal{K}, \text{ and } \eta \geq \lambda_{\ell,\pi}(x) \text{ for all } \ell \in \mathfrak{L}, \pi \in \mathcal{E}\}. \tag{A-18}$$

Next, we prove that Algorithm 1 terminates after generating finitely many feasibility cuts (7c) and optimality cuts (7d), and upon termination, it will either yield an optimal solution to (A-18) or show $\mathcal{K} = \varnothing$.

**i) Prove that there are finitely many feasibility cuts**. We first prove that at most a finite number of constraints (7c) is needed to guarantee $\hat{x} \in \mathcal{K}_2$. This implies $\hat{x} \in \mathcal{K}$, since constraints (7b) and (7e) guarantee $\hat{x} \in \mathcal{K}_1$. By the definition of $u_n(\cdot)$ (see (10)), we know that $\hat{x} \in \mathcal{K}_2$ if and only if $u_n(\hat{x}) = 0$. It follows by duality that $u_n(x) = \max \{\varphi_n^{\mathsf{T}}(h_n - T_n x) : \varphi_n \in \Phi_n\}$, for $n = 1, \ldots, N$, where

$$\Phi_n := \left\{ \varphi_n : \begin{bmatrix} W_n^{\mathsf{T}} & & \\ & I & \\ & & I \end{bmatrix} \varphi_n \leq \begin{bmatrix} 0 \\ \mathbf{1} \\ \mathbf{1} \end{bmatrix} \right\}$$

In the algorithm, if $u_n(\hat{x}) > 0$ for some $n$, Step 3 generates a feasibility cut, $\phi_n^{\mathsf{T}}(h_n - T_n x) \leq 0$. Recall that $\phi_n$ be the simplex multipliers associated with $\hat{x}$. Hence, $\phi_n$ is an extreme point of the polyhedron $\Phi_n$. The master problem (7), with this new cut added, rules out the current infeasible point in the next iteration. Hence, every generated cut is unique, and their total number is bounded by the sum of the extreme points of all $\Phi_n$ for $n = 1, \ldots, N$. Consequently, $\hat{x} \in \mathcal{K}_2$ is guaranteed after a finite number of constraints (7c) are generated. Specially, for the case that $\mathcal{K}_2 = \varnothing$, the algorithm will verify that, for any $x \in \mathcal{K}_1$, $u_n(x) > 0$ for some $n$ after finitely many iterations and then it terminates with $\mathcal{K} = \varnothing$.

**ii) Prove that there are finitely many optimality cuts**. Suppose the algorithm cannot terminate in the $k$-th iteration. Consider $\hat{x} \in \mathcal{K} \neq \varnothing$. Step 4 produces an optimality cut represented by a pair $(E_{k+1}, e_{k+1})$, i.e., $\hat{\eta} < \hat{\lambda} = e_{k+1} - E_{k+1}\hat{x}$. It means that $(E_{k+1}, e_{k+1})$ distinguishes from $(E_1, e_1), \ldots, (E_k, e_k)$. In other words, this algorithm cannot generate the same cut more than once. Notably, every cut is $\eta \geq \lambda_{\ell,\pi}(x)$ for some $\ell \in \mathfrak{L}$ and $\pi \in \mathcal{E}$. Therefore, Step 4 can generate $|\mathcal{E}| \times |\mathfrak{L}|$ cuts at most. When $\hat{\eta} \geq \hat{\lambda} = \max\{\lambda_{\ell,\pi}(\hat{x}) : \ell \in \mathfrak{L}, \pi \in \mathcal{E}\}$, the algorithm terminates and $\hat{x}$ is an optimal solution to problem (A-18).

## F. Theorems Used in the Paper

**Theorem F.1** (Theorem 1, (Rockafellar & Uryasev, 2000)). *Let $h(x, \omega)$ be a random function where $x \in \mathcal{X}$ and $\omega$ belong to an arbitrary probability space with distribution $\mathbb{Q}$. Let $q_\alpha(x)$ denote the $100(1-\alpha)$-percentile of $h(x, \omega)$ and $H_\alpha(x, t) = t + \frac{1}{\alpha} \int [h(x, \omega) - t]_+ \mathbb{Q}(d\omega)$, where $t \in \mathbb{R}$. Then, for all $x \in \mathcal{X}$, we have $\frac{1}{\alpha} \int_0^\alpha q_\tau(x)d\tau = \min_{t \in \mathbb{R}} H_\alpha(x, t)$.*

**Theorem F.2** (Theorem 2, (Athreya, 1983)). *Suppose $\liminf MN^{-\phi} > 0$ for some $\phi > 0$ as $M, N \to \infty$, and $\mathbb{E}_\mathbb{P}|F(\xi) - \mu|^\theta < \infty$ for some $\theta \geq 1$ such that $\theta\phi > 1$. Then, as $M, N \to \infty$, we have $\frac{1}{M}\sum_{m=1}^{M} F\left(\zeta_m(\hat{\mathbb{P}}_N)\right) \to 1$ w.p.1.*

**Theorem F.3** (Lemma 21.2, (van der Vaart, 1998)). *The quantile function of a cumulative distribution function $\mathcal{F}$ is the generalized inverse $\mathcal{F}^{-1} : (0, 1) \to \mathbb{R}$ given by $\mathcal{F}^{-1}(p) = \inf\{x : \mathcal{F}(x) \leq p\}$. For any any sequence of cumulative distribution functions, $\mathcal{F}_N$ converges to $\mathcal{F}$ in distribution if and only if $\mathcal{F}_N^{-1}$ converges to $\mathcal{F}^{-1}$ in distribution.*

**Theorem F.4** (Theorem 10.8 (Rockafellar, 2015)). *Let $\mathcal{C}$ be an open convex set. Let $(g_1, g_2, \cdots)$ be a sequence of finite convex functions on $\mathcal{C}$. Suppose that the sequence converges pointwise on a dense subset $\mathcal{D} \subseteq \mathcal{C}$ and the limit is finite. Then, the sequence $(g_1, g_2, \cdots)$ converges uniformly to a continuous function on any compact subset inside $\mathcal{C}$.*

**Theorem F.5** (Theorem 7.53, (Shapiro et al., 2021)). *Let $\mathcal{K}$ be a nonempty compact subset of $\mathbb{R}^n$ and suppose that (i) for any $x \in \mathcal{K}$ the function $F(\cdot, \xi)$ is continuous at $x$ for almost every $\xi \in \Xi$, (ii) $F(x, \xi)$, $x \in \mathcal{K}$, is dominated by an integrable function, and (iii) the sample is iid. Then, the expected function $f(x)$ is finite valued and continuous on $\mathcal{K}$, and the sample mean $\hat{f}_N(x)$ converges to $f(x)$ w.p.1 uniformly on $\mathcal{K}$.*

# G. Experiment Parameters

In the case study of Section 5, we consider two distinct scenarios: a regular period with probability $p$ and a worst-case period with probability $1 - p$. The worst-case scenario, which we model as a low-probability, high-impact event (like the COVID-19 pandemic), seldom occurs but has severe consequences. During the regular period, each random variable follows a uniform distribution marginally: $h_k \sim \mathcal{U}[\underline{h}_k^r, \overline{h}_k^r]$, $q_j \sim \mathcal{U}[\underline{q}_j^r, \overline{q}_j^r]$, and $w_j \sim \mathcal{U}[\underline{w}_j^r, \overline{w}_j^r]$ for $k \in \mathcal{K} := \{1, 2\}$ and $j \in \mathcal{J}$. Their joint distribution is formulated using the Gumbel copula as

$$
C(h, q, w; \lambda^r) = \exp \left\{ - \left( \sum_{k \in \mathcal{K}} \left( - \log \frac{h_k - \underline{h}_k^r}{\overline{h}_k^r - \underline{h}_k^r} \right)^{\lambda^r} + \sum_{j \in J} \left( - \log \frac{q_j - \underline{q}_j^r}{\overline{q}_j^r - \underline{q}_j^r} \right)^{\lambda^r} \right. \right.
$$

$$
\left. \left. + \sum_{j \in \mathcal{J}} \left( - \log \frac{w_j - \underline{w}_j^r}{\overline{w}_j^r - \underline{w}_j^r} \right)^{\lambda^r} \right)^{\frac{1}{\lambda^r}} \right\}
$$

Similarly, for the worst-case period, we formulate the distribution of $h$, $q$, and $w$ using a Gumbel copula with different parameters: $\underline{h}_k^w, \overline{h}_k^w, \underline{q}_j^w, \overline{q}_j^w, \underline{w}_j^w, \overline{w}_j^w$, and $\lambda^w$.

## G.1. Deterministic Parameters

1. **Unit cost (negative profit) for all products:**
   The cost vector is given by:

   $$
   c = (-14, -9, -20, -15, -4, -40, -18, -11, -13, -16, -17, -8, -9, -24, -10, -7, -12, -3, -4, -5)
   $$

2. **The amount of labor in each department required to produce components for one unit of every product:**

$$
T = \begin{bmatrix}
10 & 6 & 8 & 4 & 10 & 6 & 8 & 4 & 6 & 8 & 4 & 10 & 7 & 9 & 12 & 8 & 11 & 13 & 16 & 17 \\
6 & 2 & 3 & 2 & 6 & 2 & 3 & 2 & 3 & 2 & 6 & 2 & 5 & 3 & 7 & 4 & 6 & 5 & 8 & 9 \\
10 & 6 & 8 & 4 & 10 & 6 & 8 & 4 & 8 & 4 & 10 & 6 & 7 & 9 & 12 & 8 & 11 & 13 & 16 & 17 \\
6 & 2 & 3 & 2 & 6 & 2 & 3 & 2 & 2 & 6 & 2 & 3 & 5 & 3 & 7 & 4 & 6 & 5 & 8 & 9 \\
0 & 2 & 3 & 2 & 2 & 6 & 2 & 3 & 0 & 0 & 0 & 0 & 1 & 4 & 0 & 2 & 0 & 0 & 0 & 0 \\
0 & 0 & 0 & 0 & 0 & 0 & 0 & 10 & 6 & 8 & 4 & 0 & 0 & 0 & 0 & 0 & 0 & 9 & 0 & 0 \\
0 & 0 & 0 & 1 & 4 & 0 & 2 & 0 & 0 & 0 & 0 & 0 & 3 & 0 & 0 & 0 & 5 & 0 & 0 & 0 \\
6 & 8 & 4 & 6 & 0 & 0 & 0 & 0 & 0 & 4 & 6 & 8 & 4 & 10 & 7 & 0 & 0 & 0 & 0 & 0
\end{bmatrix}
$$

## G.2. Random Parameters

G.2.1. REGULAR PERIOD

1. $\lambda^r = 2.0$

2. Probability of regular period: $p = 0.9$

3. Show-up employee influenced by absenteeism: $h_1 \sim \mathcal{U}[\underline{h}_1^r, \overline{h}_1^r] = [8000, 8500]$

4. Temporary workers available in the future labor market. Without loss, we suppress the constraint on the capacity of total outsourced labor by setting a large value for $h_2 \sim \mathcal{U}[\underline{h}_2^r, \overline{h}_2^r] = [10000, 120000]$.

5. The labor cost of a temporary worker hired for department $j$, denoted as $\sim \mathcal{U}[\underline{q}_j^r, \overline{q}_j^r]$:

   | | | | |
   |---|---|---|---|
   | $q_1 = [3, 5]$ | $q_2 = [13, 16]$ | $q_3 = [4, 7]$ | $q_4 = [14, 17]$ |
   | $q_5 = [15, 17]$ | $q_6 = [4, 8]$ | $q_7 = [15, 19]$ | $q_8 = [18, 20]$ |

6. The efficiency of that temporary worker in department $j$, denoted as $w_j \sim \mathcal{U}[\underline{w}_j^r, \overline{w}_j^r]$:

   | | | | |
   |---|---|---|---|
   | $w_1 = [0.8, 1]$ | $w_2 = [0.8, 1]$ | $w_3 = [0.9, 1]$ | $w_4 = [0.8, 1]$ |
   | $w_5 = [0.85, 1]$ | $w_6 = [0.85, 1]$ | $w_7 = [0.9, 1]$ | $w_8 = [0.9, 1]$ |

G.2.2. WORST-CASE PERIOD

1. $\lambda^w = 5.0$

2. Probability of worst-case period: $p = 0.1$

3. Show-up employee influenced by absenteeism: $h_1 \sim \mathcal{U}[\underline{h}_1^w, \overline{h}_1^w] = [2000, 3000]$

4. Temporary workers available in the future labor market. Without loss, we suppress the constraint on the capacity of total outsourced labor by setting a large value for $h_2 \sim \mathcal{U}[\underline{h}_2^w, \overline{h}_2^w] = [10000, 120000]$.

5. The labor cost of a temporary worker hired for department $j$, denoted as $q_j \sim \mathcal{U}[\underline{q}_j^w, \overline{q}_j^w]$:

$$q_1 = [9, 12] \qquad q_2 = [21, 25] \qquad q_3 = [10, 12] \qquad q_4 = [22, 24]$$
$$q_5 = [18, 20] \qquad q_6 = [18, 21] \qquad q_7 = [18, 20] \qquad q_8 = [22, 25]$$

6. The efficiency of that temporary worker in department $j$, denoted as $w_j \sim \mathcal{U}[\underline{w}_j^w, \overline{w}_j^w]$:

$$w_1 = [0.5, 0.6] \qquad w_2 = [0.5, 0.6] \qquad w_3 = [0.6, 0.7] \qquad w_4 = [0.4, 0.6]$$
$$w_5 = [0.55, 0.65] \qquad w_6 = [0.55, 0.65] \qquad w_7 = [0.6, 0.7] \qquad w_8 = [0.6, 0.7]$$

## G.3. Parameters for the fixed-recourse product-mix problem

Section 5.3 adapts a two-stage product-mix problem with fixed recourse (King, 1988).

$$A = 0, \quad b = 0, \quad c = [-12, -20, -18, -40]^\mathsf{T},$$
$$q = [6, 12, 0, 0]^\mathsf{T}, \quad h = [500\gamma_1, 500\gamma_2]^\mathsf{T},$$
$$T = \begin{bmatrix} 4 - \frac{\gamma_1}{4} & 9 - \frac{\gamma_1}{4} & 7 - \frac{\gamma_1}{4} & 10 - \frac{\gamma_1}{4} \\ 3 - \frac{\gamma_2}{4} & 1 - \frac{\gamma_2}{4} & 3 - \frac{\gamma_2}{4} & 6 - \frac{\gamma_2}{4} \end{bmatrix}, \qquad W = \begin{bmatrix} -0.9 & 0 & 1 & 0 \\ 0 & -0.9 & 0 & 1 \end{bmatrix},$$

where

$$[\gamma_1, \gamma_2]^\mathsf{T} \sim \frac{7}{10} \mathcal{N}\left( \begin{bmatrix} 12 \\ 8 \end{bmatrix}, \begin{bmatrix} 5.76 & 1.92 \\ 1.92 & 2.56 \end{bmatrix} \right) + \frac{3}{10} \mathcal{N}\left( \begin{bmatrix} 2 \\ 1 \end{bmatrix}, \begin{bmatrix} 0.16 & 0.04 \\ 0.04 & 0.04 \end{bmatrix} \right).$$

## G.4. Parameters for the newsvendor problem

The newsvendor experiment in Section D uses

$$F(x, \xi) = p^\mathsf{T} x + h^\mathsf{T}(x - \xi)_+ + b^\mathsf{T}(\xi - x)_+.$$

Here $x$ is the order vector, $\xi$ is the random demand vector, and $p$, $h$, and $b$ are the unit purchase, overage, and underage costs.

$$p = -2, \qquad h = 9, \qquad b = 5.$$
$$\mu_1 = [60.89, 48.58, 46.81, 56.54, 61.58, 52.69, 69.42, 60.54, 54.43, 51.76]^\mathsf{T}$$
$$\mu_2 = [50.30, 61.87, 53.16, 41.79, 51.94, 62.14, 45.47, 45.26, 55.95, 55.95]^\mathsf{T}$$

$$\Sigma_1 = \begin{bmatrix} 9.27 & 2.84 & -0.07 & 1.19 & -0.48 & 1.40 & 2.87 & 4.06 & -1.40 & -1.96 \\ 2.84 & 5.90 & -2.83 & 0.21 & 2.27 & -2.40 & -0.89 & 4.22 & 3.43 & 2.78 \\ -0.07 & -2.83 & 5.48 & -0.30 & 0.90 & 3.54 & -4.51 & -2.45 & -2.91 & -4.95 \\ 1.19 & 0.21 & -0.30 & 7.99 & -1.02 & -1.27 & -0.15 & -1.55 & -1.69 & -0.36 \\ -0.48 & 2.27 & 0.90 & -1.02 & 9.48 & -0.08 & -3.69 & 2.71 & -0.69 & -0.34 \\ 1.40 & -2.40 & 3.54 & -1.27 & -0.08 & 6.94 & -1.26 & -2.73 & 0.01 & -5.19 \\ 2.87 & -0.89 & -4.51 & -0.15 & -3.69 & -1.26 & 12.05 & -0.16 & -0.16 & 2.44 \\ 4.06 & 4.22 & -2.45 & -1.55 & 2.71 & -2.73 & -0.16 & 9.16 & -0.77 & 1.94 \\ -1.40 & 3.43 & -2.91 & -1.69 & -0.69 & 0.01 & -0.16 & -0.77 & 7.41 & 2.24 \\ -1.96 & 2.78 & -4.95 & -0.36 & -0.34 & -5.19 & 2.44 & 1.94 & 2.24 & 6.70 \end{bmatrix}$$

$$\Sigma_2 = \begin{bmatrix} 6.32 & 2.99 & -0.06 & 0.73 & -0.33 & 1.36 & 1.55 & 2.51 & -1.19 & -1.75 \\ 2.99 & 9.57 & -4.09 & 0.19 & 2.44 & -3.60 & -0.74 & 4.02 & 4.49 & 3.83 \\ -0.06 & -4.09 & 7.06 & -0.25 & 0.86 & 4.74 & -3.35 & -2.08 & -3.40 & -6.08 \\ 0.73 & 0.19 & -0.25 & 4.37 & -0.64 & -1.11 & -0.07 & -0.86 & -1.29 & -0.29 \\ -0.33 & 2.44 & 0.86 & -0.64 & 6.74 & -0.08 & -2.04 & 1.71 & -0.60 & -0.31 \\ 1.36 & -3.60 & 4.74 & -1.11 & -0.08 & 9.65 & -0.98 & -2.41 & 0.01 & -6.62 \\ 1.55 & -0.74 & -3.35 & -0.07 & -2.04 & -0.98 & 5.17 & -0.08 & -0.10 & 1.72 \\ 2.51 & 4.02 & -2.08 & -0.86 & 1.71 & -2.41 & -0.08 & 5.12 & -0.59 & 1.57 \\ -1.19 & 4.49 & -3.40 & -1.29 & -0.60 & 0.01 & -0.10 & -0.59 & 7.83 & 2.49 \\ -1.75 & 3.83 & -6.08 & -0.29 & -0.31 & -6.62 & 1.72 & 1.57 & 2.49 & 7.83 \end{bmatrix}$$

$$\varepsilon \sim \begin{bmatrix} \mathcal{U}(-5.37, 26.27) \\ \mathcal{U}(6.74, 14.16) \\ \mathcal{U}(3.22, 17.68) \\ \mathcal{U}(-7.48, 28.38) \\ \mathcal{U}(-4.89, 25.79) \\ \mathcal{U}(-0.21, 16.11) \\ \mathcal{U}(-12.14, 32.99) \\ \mathcal{U}(-7.74, 28.64) \\ \mathcal{U}(0.77, 20.13) \\ \mathcal{U}(2.13, 18.77) \end{bmatrix}.$$

