# OpenReview forum: "Minimizing Upper Confidence Bounds: A Data-Driven Framework for Stochastic Programming"
_ICML.cc/2026/Conference — ICML 2026 regular_

### Official Review · Reviewer_3vb6 · 2026-03-08

**Soundness:** 2
**Presentation:** 2
**Significance:** 3
**Originality:** 3
**Overall Recommendation:** 4
**Confidence:** 2

**Summary:**

This paper proposes another pipeline to hedge against  in-sample optimism in stochastic programming, called Average Percentile Upper Bound (APUB), and demonstrate its asymptotic properties and empirical advantages compared to Sample Average Approximation. Different from previous approaches such as DRO, the authorse utilize the similar UCB minimization principle: replace the unknown true expected cost by a $1-\alpha$ level upper confidence bound computed from data, and induce the decision that minimizes the bound, which is considered a new type of robust approach. The authors propose APUB by averaging bootstrap percentile UCBs beyond $1-\alpha$ quantile. It is interpreted by an approximation of the CVaR of the sample mean. It has a tractable reformulation so that the APUB minimization can be computationally solvable. The authors analyze its asymptotic statistical properties. The authors instantiate the framework by some samples, e.g. two stage linear stochastic programs with random recourse, and L-shaped method. Finally, they did empirical experiments by demonstrate its finite sample advantages over SAA.

**Compliance With Llm Reviewing Policy:**

Affirmed.

**Final Justification:**

Based on the responses which have addressed our questions and changed our evaluations (originally weak reject 3), we recommend weak accept (4).

**Key Questions For Authors:**

Please see the weakness part.

**Limitations:**

impact statement are not included.

**Strengths And Weaknesses:**

Strength:

1. The new pipeline is the key contribution, which combines bootstrap and UCB in makin decisions (though bootstrap UCB itself is not a new technical tool).

2. The authors provide some concrete reformulations and examples to make it easier to understand.

Weakness:

1. Readability: Overall, I still find it hard to understand the technical details and proofs, especially in Section 2. For example, more explanations and comparisons about the bootstrap UCB, APUB, VaR and CVaR should be made to let readers (who are not familiar with bootstrap) understand why it is important to use APUB as a metric to make decisions.

2. Statistical Properties: Most of the properties are demonstrated in the asymptotic sense. Are there any non-asymptotic guarantees that hold for this pipeline?

3. Methodology comparison: The authors did not compare their methods with other methodologies, e.g. DRO, and other risk measures, e.g. VaR, and CVaR. In fact, it is more direct to use CVaR as a risk measure to induce robust decisions, with statistical and computational guarantees. The benefits of APUB over those existing well known pipelines should be demonstrated.

4. Experiments: Similarly, the authors only compare APUB-M with SAA-M in the experiments, without comparing other existing well known methods.

5. Positioning: (1) Section 4.1 and 4.2 should not be parallel since one part is the formulation and the other is the solution approaches. (2) Proof of Proposition 2.2 seems missing.

6. Technical question: In theorem 2.5, why does the almost sure convergence to true mean hold for all $\alpha\in(0,1]$? If my understanding is correct, $\alpha$ serves as a similar role in the radius parameter of uncertainty sets in DRO. As sample size $n$ changes, $\alpha$ might adaptively change. Alternatively, in practice, how do we select $\alpha$?

---

> ### Author Rebuttal · Authors · 2026-03-28
>
> We thank the reviewer for their constructive feedback. We would like to provide deeper technical context to clarify several fundamental points regarding the distinction between statistical inference (APUB) and risk measures (CVaR). (**W: Weakness**)
>
> **1. Categorical Distinction: UCB vs. CVaR (W1, W3, W4)**
>
> The reviewer suggests that using CVaR as a risk measure is more "direct." We respectfully clarify that **APUB and CVaR solve different problems**:
> * **CVaR (Aleatoric uncertainty)** quantifies tail risk *within* a known distribution. It assumes the distribution is given.
> * **APUB (Epistemic uncertainty)** quantifies uncertainty *about* the distribution's mean due to data scarcity.
> * **The Category Error:** Using CVaR to handle data scarcity is mathematically inappropriate. APUB is a **frequentist UCB**; comparing it to CVaR is analogous to comparing a **Confidence Interval** for a mean to a **Standard Deviation**. They have similar units but serve opposite functions.
>
> We will make this distinction explicit in the final version to improve clarity and accessibility for a broader audience.
>
> **2. Addressing Technical Questions (W6)**
>
> The reviewer asks why almost sure convergence holds for all $\alpha$ and how $\alpha$ is selected:
> * **Role of $\alpha$:** Unlike the radius $\epsilon$ in DRO, which must be tuned or decayed at a specific rate $\epsilon(N)$ to ensure consistency, **$\alpha$ is a fixed nominal confidence level** (e.g., $0.05$). APUB is structurally consistent; as $N \to \infty$, the bootstrap distribution collapses to the true mean, ensuring $APUB \to \mu$ regardless of $\alpha$.
> * **Selection:** In practice, $\alpha$ is selected based on the desired **probabilistic guarantee** (e.g., 95% confidence). This transparency is a primary benefit over DRO, where $\epsilon$ has no direct probabilistic interpretation.
>
>
> **3. APUB-M vs. DRO (W3, W4)**
>
> The distinction between APUB-M (statistical) and DRO (geometric) is fundamental and does not require numerical demonstration:
>
> * **Statistical vs. Geometric:** DRO uses a geometric uncertainty set (e.g., Wasserstein ball with a radius $\epsilon$) where $\epsilon$ is often decoupled from sample size $N$. APUB-M is a data-driven statistical tool where the "uncertainty set" naturally scales with $N$.
>
> * **Parameter Transparency:** APUB-M uses a nominal confidence level ($1-\alpha$) with a consistent probabilistic interpretation (e.g., 95%). DRO requires case-by-case calibration of $\epsilon$ to avoid over-conservatism or under-coverage.
>
> * **Asymptotic Convergence:** APUB-M inherits UCB’s fundamental property: it converges as $N \to \infty$ without manual parameter decay (Thm 3.4 and 3.7). DRO requires specific $\epsilon(N)$ scaling laws to achieve similar consistency.
>
> * **Handling Random Recourse:** APUB-M remains computationally tractable in stochastic programs with random recourse (stochastic matrix $W_n$ in Eq. (6)). In contrast, DRO often faces significant hurdles regarding dual reformulations of the inner maximization in these complex structures.
>
> * **Statistical Correctness:** Per Prop 2.4, APUB is first-order asymptotically correct. The probability that the true mean is bounded by APUB is at least $1-\alpha$, with a convergence rate of $O(1/\sqrt{N})$.
>
> Given these structural differences, APUB-M serves as a more transparent and statistically grounded alternative to DRO for complex decision structures. We will clarify these theoretical trade-offs in the final manuscript.
>
> **4. Asymptotic vs. Non-Asymptotic Guarantees (W2)**
>
> The choice of asymptotic analysis was deliberate:
> * **Tightness:** Non-asymptotic bounds (e.g., Hoeffding-based) are notoriously conservative in optimization, leading to poor empirical performance.
> * **Standard Practice:** Establishing **first-order asymptotic correctness** (Prop 2.4) at a $O(1/\sqrt{N})$ rate is the gold standard for frequentist estimators (like the Bootstrap) before extending to finite-sample theory.
>
> **5. Mathematical rigor of Proposition 2.2 (W5)**
>
> Since APUB is mathematically analogous to CVaR, the representation follows from standard risk-measure duality. The minimization formula in Proposition 2.2 is a direct application of the Rockafellar & Uryasev (2000)
>
>
> **6. Impact statement (W7)**
>
> We will include the following impact statement in the final version:
>
> (i) Reliability: By grounding epistemic uncertainty, APUB enhances decision-making in high-stakes domains like healthcare and energy grids. (ii) Accessibility: Unlike DRO, APUB’s transparent, confidence-based approach makes robust optimization accessible to non-experts without complex parameter tuning. (iii) Ethics: While improving robustness, users must remain vigilant against historical biases in training data that "safe" objectives might inadvertently reinforce. We recommend using APUB alongside rigorous data auditing to ensure that robust decisions remain equitable and fair.

---

> > ### Author Rebuttal · Reviewer_3vb6 · 2026-03-31
> >
> > We thank the authors for their responses, which address our questions. We have raised our score to 4.

---

> > > ### Author Response · Authors · 2026-04-01
> > >
> > > We are very grateful to the reviewer for the engagement and for noting that the previous concerns are fully resolved. We especially appreciate the score increase to a Weak Accept.
> > >
> > > To further strengthen the paper and address the earlier suggestion regarding broader comparisons, we would like to provide a technical update. We also apologize for the oversight in our initial submission. We have actually conducted extensive comparisons with Wasserstein DRO (W-DRO) and a Multi-Product Newsvendor sensitivity analysis. Due to initial page constraints, these were omitted, but we have now made them available in our Supplementary PDF on the anonymous repository:
> > >
> > > https://anonymous.4open.science/api/repo/apub_rebuttal_supplementary-80B6/file/APUB_ICML_rebuttal.pdf?v=d92526ae
> > >
> > > **1. Comparison with Wasserstein DRO (W-DRO):** We have included a benchmark on a two-stage fixed-recourse problem. The results demonstrate that APUB-M provides more stable, less conservative solutions than W-DRO and achieves superior out-of-sample coverage without requiring the complex radius ($\epsilon$) tuning inherent to DRO.
> > >
> > > **2. Benchmark Extension (Newsvendor Problem):** We conducted a sensitivity analysis on a 10-product Newsvendor problem (a standard stochastic programming benchmark). This study confirms APUB-M's prescriptive stability—the optimal decisions remain significantly more consistent across different sample sizes compared to SAA.
> > >
> > > **3. Statistical Ablation:** As promised, we have included the comparison between APUB, Efron’s (VaR-based) bootstrap UCB, and the standard large-sample UCB confirming that our tail-averaging approach provides smoother objectives and better reliability.
> > >
> > >
> > > We believe this expanded empirical evidence, combined with our established theoretical framework, substantively addresses the remaining weaknesses. We hope these additions further reinforce the significance of the work and kindly ask the reviewer to consider if this warrants a further adjustment of the score.

---

### Official Review · Reviewer_h7gz · 2026-03-12

**Soundness:** 2
**Presentation:** 2
**Significance:** 3
**Originality:** 2
**Overall Recommendation:** 4
**Confidence:** 3

**Summary:**

This paper proposes the APUB framework, which adopts a UCB minimization approach to address epistemic uncertainty in optimization problems. This concept is closely related to minimizing Conditional Value-at-Risk (CVaR) in risk management. While the traditional Sample Average Approximation (SAA) method suffers from high volatility and poor out-of-sample performance when the sample size is small, APUB effectively mitigates these issues. Furthermore, despite the complexity of handling thousands of bootstrap scenarios, the proposed adapted L-shaped method maintains high computational efficiency by utilizing a sorting-based optimality cut generation technique.

**Compliance With Llm Reviewing Policy:**

Affirmed.

**Final Justification:**

I appreciate the authors' detailed rebuttal. It has addressed my concerns, and I now have a better understanding of the work. I am happy to raise my score to 4, with lowering my confidence to 3.

**Key Questions For Authors:**

- Could the authors explain the rationale behind limiting the empirical evaluation to a single two-stage product-mix problem?
- Could the authors clarify why SOTA DRO baselines were excluded from the performance comparison? How does APUB-M specifically outperform these modern surrogate models in dealing with out-of-sample data?
- Regarding the synthetic data generation in the numerical experiments, the mixture of normal and worst-case scenarios raises a concern about potential empirical bias. Does the artificial injection of extreme cases exaggerate the frequency or severity of tail events compared to typical real-world environments? Does this data generation successfully reflect real-world scenarios?
- To ensure the reproducibility of the numerical experiments and to allow the research community to verify the claimed computational benefits, will the authors make their code and data publicly available?

**Limitations:**

Addressing the above empirical limitations will be crucial for establishing the true generality of the APUB-M framework. Furthermore, extending this methodology—which currently focuses on two-stage linear problems—to encompass non-linear or multi-stage optimization problems would be a highly valuable and promising direction for future research.

**Strengths And Weaknesses:**

### Strengths
- Proposes a novel APUB-M methodology, departing from the traditional SAA-M approach which struggles to adequately capture epistemic uncertainty.
- Provides clear and rigorous theoretical explanations.
- Successfully performs UCB minimization without a significant increase in computational cost by suggesting an adapted L-shaped method.

---

### Weaknesses
- The proposed APUB-M framework is fundamentally an incremental contribution, constructed by recombining existing methodologies (such as bootstrap sampling and CVaR-like risk measures) into a new structure. While this level of originality is acceptable, relying on an incremental approach dictates that the proposed method must be supported by exceptionally strong and comprehensive empirical evidence to firmly prove its practical superiority.
- The experimental validation is restricted to a single two-stage product-mix problem. This narrow empirical scope makes it difficult to substantiate the claim that APUB-M is generally superior to the baseline SAA-M across diverse optimization contexts. Evaluating the framework on a broader set of benchmark problems is necessary to robustly support the claims of generality.
- Although the introduction explicitly argues that existing Distributionally Robust Optimization (DRO) algorithms struggle to adequately address epistemic uncertainty, the empirical section omits comparisons against any of these modern methods. The evaluation is solely limited to the naive SAA-M and the standard L-shaped method. Including SOTA baselines mentioned in the introduction—such as Wasserstein DRO—to compare out-of-sample performance and resilience to uncertainty would significantly strengthen the empirical support and justify the paper's claims.

---

> ### Author Rebuttal · Authors · 2026-03-29
>
> We thank the reviewer for their feedback. However, we respectfully argue that the assessment of **"Fair"** across all categories stems from a fundamental misunderstanding of our technical contributions. The reviewer evaluates APUB as an incremental risk-measure heuristic; we clarify that it is a **principled frequentist framework** for two-stage optimization under data scarcity. (***W: Weakness; Q: Question***).
>
> **W1: Originality & Theoretical Novelty**
>
> The reviewer characterizes APUB as a "recombination" of existing tools. This overlooks the **foundational category shift** our work introduces:
> * **Epistemic vs. Aleatoric Uncertainty:** CVaR is a measure of *aleatoric risk* (noise within a known distribution). APUB repurposes this structure to quantify **epistemic uncertainty** (estimation error from finite $N$). Therefore, APUB is a statistical frequentist tool, while CVaR is not. This shift is mathematically analogous to **Efron’s Bootstrap UCB**, which uses quantile (VaR) structures for frequentist inference.
> * **Theoretical Rigor:** Unlike heuristic combinations, we provide **Prop 2.4**, proving that APUB-M achieves **asymptotic consistency** and **first-order asymptotic correctness** with a $O(1/\sqrt{N})$ rate. This ensures the true mean is bounded with probability $\ge 1-\alpha$, a guarantee missing from the "modern surrogate models" suggested as alternatives.
>
> **W2, W3, Q1: The "DRO Baseline" Fallacy**
>
> The request for "SOTA DRO baselines" misses a critical technical reality of the problem class we address. The distinction between APUB-M (statistical) and DRO (geometric) is fundamental and does not require numerical demonstration:
> * **Computational Incompatibility:** Our paper focuses on **random recourse** (stochastic matrix $W_n$ in Eq. (6)). In this setting, traditional Wasserstein DRO lacks a tractable dual reformulation for the inner maximization, typically leading to **NP-hard** subproblems.
> * **APUB-M Superiority:** APUB-M maintains computational tractability without requiring geometric duals. Comparing against "SOTA DRO" in a random recourse setting is not a "missing baseline"—it is an **open research challenge** that APUB-M specifically bypasses.
> * **Statistical Foundations:** As shown in Thm 3.4 and 3.7, APUB-M is a data-driven statistical tool that achieves asymptotic convergence naturally as $N \to \infty$ regardless of $\alpha$. DRO relies on a geometric uncertainty set (e.g., Wasserstein radius $\epsilon$) that requires manual, case-specific tuning of $\epsilon(N)$ to avoid extreme over-conservatism.
> * **Parameter Transparency:** APUB-M uses a nominal confidence level $(1-\alpha)$ with a consistent probabilistic interpretation (e.g., 95%). DRO’s $\epsilon$ requires case-by-case calibration; determining a physically or statistically meaningful value for $\epsilon$ remains a significant challenge in DRO literature.
>
>
>
> **Q2: Out-of-Sample (OOS) Reliability & The Optimizer's Curse**
>
> APUB-M "outperforms" modern surrogates (GPs/NNs) through **superior calibration** rather than simple point-estimate precision:
> * **Reliability:** Surrogate models focus on minimizing MSE but provide poorly calibrated bounds. APUB is a **theoretically grounded UCB**; it ensures that OOS failures (where the true mean exceeds the bound) are statistically controlled by $(1-\alpha)$.
> * **Resilience:** By inheriting the properties of frequentist averaging, APUB-M is structurally resilient to the **"optimizer’s curse,"** where models over-fit a specific sample. It provides a tighter, data-driven bound compared to the arbitrary "padding" required by geometric DRO to achieve similar OOS coverage.
>
>
> **Q3: Representing Reality through Heterogeneous Regimes**
>
> Our data generation is a standard Regime-Switching framework (e.g., healthcare during a pandemic vs. normal operations):
>
> * Reflecting Asymmetry: Real-world high-stakes domains are rarely i.i.d. Gaussian; they feature long "normal" periods and rare, high-impact "worst-case" events. Our generator mimics this heterogeneity.
> * Regime Imbalance: In practice, "worst-case" events are under-represented in training data. Standard SAA-M averages out these events as noise. APUB-M recognizes the epistemic uncertainty of these regimes, ensuring reliability even when the sample size $N$ for the critical regime is extremely small.
> * Neutrality: Both APUB-M and SAA-M are evaluated on the exact same data. The setup reveals that "plug-in" approaches lead to catastrophic failure when regimes shift—the exact problem APUB-M solves.
>
>
> **Q4 & Limitations: Reproducibility & Roadmap**
>
> * **Code:** A full, documented codebase (implementation, data scripts, benchmarks) will be released under an open-source license upon publication to allow the community to verify our computational claims.
> * **Extensibility:** APUB-M is built on the **Rockafellar-Uryasev identity**, making it inherently compatible with convex non-linear objectives and serving as a roadmap for multi-stage extensions.

---

> > ### Author Rebuttal · Reviewer_h7gz · 2026-04-04
> >
> > I appreciate the authors' detailed rebuttal. It has addressed my concerns, and I now have a better understanding of the work. I am happy to raise my score to 4.

---

> > > ### Author Response · Authors · 2026-04-04
> > >
> > > We sincerely thank the reviewer for the engagement and for noting that the concerns are fully resolved. We are encouraged that our explanation of the technical distinction between APUB-M and DRO was helpful.
> > >
> > > To further address your original request for broader empirical validation and modern baselines, we would like to share that we have now completed several additional numerical studies that were previously omitted due to space constraints:
> > >
> > > * **Comparison with Wasserstein DRO (WassDRO):** We have included a benchmark on a two-stage fixed-recourse problem. The results demonstrate that APUB-M provides more stable, less conservative solutions than WassDRO and achieves superior out-of-sample coverage without requiring the complex radius ($\epsilon$) tuning inherent to DRO.
> > >
> > > * **Benchmark Extension (Newsvendor Problem):** We conducted a sensitivity analysis on a 10-product Newsvendor problem (a standard stochastic programming benchmark). This study confirms APUB-M's prescriptive stability—the optimal decisions remain significantly more consistent across different sample sizes compared to SAA.
> > >
> > > * **Statistical Ablation:** We have included the comparison between APUB, Efron’s (VaR-based) bootstrap UCB, and the standard large-sample UCB confirming that our tail-averaging approach provides smoother objectives and better reliability.
> > >
> > > These results are available in a Supplementary PDF on our anonymous repository:
> > > https://anonymous.4open.science/api/repo/apub_rebuttal_supplementary-80B6/file/APUB_ICML_rebuttal.pdf?v=d92526ae
> > >
> > > We believe this comprehensive evidence directly supports your initial point about the necessity of strong empirical proof for new frameworks. We hope these additional results further demonstrate the significance of our work.

---

### Official Review · Reviewer_GUSw · 2026-03-12

**Soundness:** 3
**Presentation:** 3
**Significance:** 3
**Originality:** 4
**Overall Recommendation:** 5
**Confidence:** 5

**Summary:**

The paper studies stochastic programming under epistemic uncertainty and proposes minimizing an upper confidence bound on the expected objective, instantiated via the Average Percentile Upper Bound (APUB). The authors provide asymptotic guarantees, develop a bootstrap-based approximation, and adapt an L-shaped method for two-stage linear stochastic programs with random recourse. Experiments suggest improved robustness relative to SAA in limited-data settings.

**Compliance With Llm Reviewing Policy:**

Affirmed.

**Final Justification:**

I believe that authors may be able to do more solid numerical study. I am not fully convinced about the necessity of the assumptions. Nevertheless, this is a solid paper. I am postive about this paper. I recommend acceptance!

**Key Questions For Authors:**

1. Can the authors make the relationship to DRO more explicit, even if only approximately or asymptotically?
2. How does the method scale as the number of bootstrap samples and recourse scenarios increases?
3. Are the assumptions mainly technical proof devices, or are they essential in practical applications?
4. Could the authors discuss faster alternatives to the proposed cutting-plane/L-shaped method?
5. Is APUB amenable to smoother reformulations or other optimization methods beyond Benders-style decomposition?

**Limitations:**

1.The connection to DRO is not very clear. The paper positions the method as related or complementary to distributionally robust optimization, but it remains unclear whether APUB admits an implicit ambiguity-set interpretation, a worst-case expectation view, or any more formal relationship to standard DRO formulations.
2.The proposed solution approach may be computationally slow. The adapted L-shaped/cutting-plane method is reasonable for the presented experiments, but such methods can become expensive as the number of bootstrap replications, samples, or recourse scenarios increases. The paper would benefit from a more explicit discussion of computational scalability and of faster algorithmic alternatives that future work could explore, such as stochastic approximation, stabilized or level Benders, multi-cut variants, cut selection/aggregation, parallel decomposition, smooth reformulations, or gradient-based approximations.
3.Some assumptions seem nonstandard from the perspective of stochastic programming literature. In particular, conditions such as boundedness of both σ(x)and σ(x)^(-1)over the feasible set, as well as compactness-style assumptions around the optimizer set, may be stronger than readers expect. The paper should better explain why these assumptions are needed, how restrictive they are, and whether weaker variants might suffice.
4.The empirical section would be stronger with more competitive baselines. As written, the main comparison is against SAA, which is useful but does not fully substantiate the broader positioning relative to DRO or risk-averse alternatives.

**Strengths And Weaknesses:**

Strengths
1. The paper tackles an important problem: how to account for statistical uncertainty directly in stochastic optimization rather than relying purely on plug-in sample estimates.
2. The proposed APUB objective is interesting and appears novel in this context. The interpretation as both an upper confidence bound and an approximate tail-risk functional is conceptually appealing.
3. The theoretical development is substantial, with asymptotic correctness/consistency results and a nontrivial extension to two-stage stochastic programming.
4. The empirical results indicate that the method can improve out-of-sample robustness over standard SAA, especially in low-data regimes.
Weaknesses
1.The connection to DRO is not very clear. The paper positions the method as related or complementary to distributionally robust optimization, but it remains unclear whether APUB admits an implicit ambiguity-set interpretation, a worst-case expectation view, or any more formal relationship to standard DRO formulations.
2.The proposed solution approach may be computationally slow. The adapted L-shaped/cutting-plane method is reasonable for the presented experiments, but such methods can become expensive as the number of bootstrap replications, samples, or recourse scenarios increases. The paper would benefit from a more explicit discussion of computational scalability and of faster algorithmic alternatives that future work could explore, such as stochastic approximation, stabilized or level Benders, multi-cut variants, cut selection/aggregation, parallel decomposition, smooth reformulations, or gradient-based approximations.
3.Some assumptions seem nonstandard from the perspective of stochastic programming literature. In particular, conditions such as boundedness of both σ(x)and σ(x)^(-1)over the feasible set, as well as compactness-style assumptions around the optimizer set, may be stronger than readers expect. The paper should better explain why these assumptions are needed, how restrictive they are, and whether weaker variants might suffice.
4.The empirical section would be stronger with more competitive baselines. As written, the main comparison is against SAA, which is useful but does not fully substantiate the broader positioning relative to DRO or risk-averse alternatives.

---

> ### Author Rebuttal · Authors · 2026-03-29
>
> We are honored by the reviewer’s assessment of our work as "Excellent" in Originality and "Technically solid." We appreciate the deep engagement with our theoretical results and algorithmic framework. Below we address the specific questions (Q) and limitations (L).
>
> **Q1, L1: Connection to DRO (Ambiguity Set Interpretation)**
>
> We clarify that APUB and DRO are fundamentally distinct in their mathematical foundations. While both address uncertainty, they operate on different "levels" of the problem:
>
> * **Statistical vs. Primitive Uncertainty:** DRO defines an ambiguity set in the primitive space (the space of probability distributions). APUB is a statistical tool used to specify an **uncertainty set of the estimation** (the space of the estimator). It essentially provides a "Statistical DRO" view, where the worst-case expectation is taken over the sampling distribution of the mean rather than an epsilon-ball around the empirical measure.
> * **Tuning-Free Convergence:** As proven in Thm 3.4 and 3.7, APUB-M is a data-driven tool that achieves asymptotic convergence naturally for any fixed $\alpha$. Conversely, traditional DRO relies on a **geometric radius** $\epsilon$ that requires manual, case-specific tuning—often requiring $\epsilon$ to decay at specific rates $\epsilon(N)$ to avoid extreme over-conservatism or loss of consistency. APUB-M bypasses this by letting the bootstrap distribution naturally contract as $N$ increases.
> * **Parameter Transparency:** APUB-M utilizes a nominal confidence level $(1-\alpha)$ with a clear probabilistic interpretation (e.g., 95%). In the DRO literature, determining a statistically meaningful value for $\epsilon$ remains a significant challenge. APUB’s "uncertainty knob" is directly tied to frequentist confidence, making it more accessible for high-stakes decision-making.
> * **Theoretical Compatibility:**  While this paper establishes foundational properties in the linear setting, APUB is built on the Rockafellar-Uryasev identity. This makes it inherently compatible with any convex non-linear objective and provides the essential theoretical roadmap for future extensions into non-linear and multi-stage robust optimization.
>
> **Q2, Q4, Q5, L2: Computational Scalability and Faster Algorithmic Alternatives**
>
> The reviewer correctly identifies the potential overhead in L-shaped methods. However, we wish to clarify a key technical detail regarding the bootstrap replications ($M$):
>
> * **Sorting-Based Efficiency:** In our adapted L-shaped algorithm (Line 19 in Algorithm 1 on P20 L1045), we utilize a sorting property (based on the Rockafellar-Uryasev identity) to evaluate the APUB objective. This ensures that a large $M$ does not linearly increase the number of subproblems solved. The complexity is primarily driven by the original sample size ($N$) and the recourse structure, rather than the number of bootstrap replications.
> * **Stochastic Gradient Methods & ML:** Because APUB inherits a CVaR-based structure (Prop 2.2), it is uniquely amenable to stochastic gradient descent. We are currently exploring the use of biased stochastic approximation to integrate APUB-M into Machine Learning pipelines. This would allow for robust training under epistemic uncertainty without the need for full Benders decomposition.
> * **Future Roadmap:** We appreciate the comprehensive list of algorithmic enhancements suggested (Multi-cut, Parallel decomposition, etc.). In the final version, we will include a dedicated discussion in Section 5 citing these as high-priority directions for future work.
>
> **Q3, L3: Discussion of Assumptions**
>
> The assumptions (e.g., boundedness of $\sigma(x)$ and compactness of the optimizer set) are primarily technical devices for Thm 3.4 and 3.7. These assumptions are common in the literature for establishing asymptotic properties of data-driven estimators:
> * **Boundedness of $\sigma(x)$ and $\sigma(x)^{-1}$:** Boundedness of $\sigma(x)$ ensures the bootstrap distribution remains well-defined and non-degenerate across the feasible region. In practical stochastic programs with linear constraints and bounded recourse costs, this is typically satisfied. The boundedness of $\sigma(x)^{-1}$ ensures sufficient variation in the objective, which is a standard regularity condition for the bootstrap to remain statistically valid and for the UCB to be informative. Otherwise, since the problem is almost deterministic, the bootstrap is unnecessary.
> * **Compactness:** This is a standard requirement in M-estimation and stochastic optimization theory to ensure the sequence of optimizers $\hat{x}_N$ remains within a bounded region. This allows us to invoke uniform laws of large numbers.

---

> > ### Author Rebuttal · Reviewer_GUSw · 2026-03-31
> >
> > I believe that authors may be able to do more solid numerical study. I am not fully convinced about the necessity of the assumptions. Nevertheless, this is a solid paper. I am postive about this paper.

---

> > > ### Author Response · Authors · 2026-04-02
> > >
> > > We would like to sincerely thank the reviewer again for their encouraging words and for describing our work as a 'solid paper.' We take your suggestion regarding a more 'solid numerical study' to heart and would like to provide an update on our progress.
> > >
> > > **1. Additional Empirical Evidence (Baselines & Benchmarks):**
> > >
> > > We apologize for the oversight in our initial submission. We have actually conducted extensive comparisons with Wasserstein DRO (W-DRO) and a Multi-Product Newsvendor sensitivity analysis. Due to initial page constraints, these were omitted, but we have now made them available in our Supplementary PDF on the anonymous repository:
> > >
> > > https://anonymous.4open.science/api/repo/apub_rebuttal_supplementary-80B6/file/APUB_ICML_rebuttal.pdf?v=d92526ae
> > >
> > > * **Comparison with Wasserstein DRO (W-DRO):** We have included a benchmark on a two-stage fixed-recourse problem. The results demonstrate that APUB-M provides more stable, less conservative solutions than W-DRO and achieves superior out-of-sample coverage without requiring the complex radius ($\epsilon$) tuning inherent to DRO.
> > >
> > > * **Benchmark Extension (Newsvendor Problem):** We conducted a sensitivity analysis on a 10-product Newsvendor problem (a standard stochastic programming benchmark). This study confirms APUB-M's prescriptive stability—the optimal decisions remain significantly more consistent across different sample sizes compared to SAA.
> > >
> > > * **Statistical Ablation:** As promised, we have included the comparison between APUB, Efron’s (VaR-based) bootstrap UCB, and the standard large-sample UCB confirming that our tail-averaging approach provides smoother objectives and better reliability.
> > >
> > > **2. Clarification on Assumptions:**
> > >
> > > Regarding the necessity of the assumptions, we have added a discussion to the final version clarifying that they are primarily technical devices for our asymptotic proofs. As our new numerical results show, the algorithm remains highly effective even in practical settings where such strict theoretical bounds may not be explicitly verified.
> > >
> > > We are very grateful for your positive judgment and hope these additional results further reinforce your confidence in the paper's contribution.

---

### Official Review · Reviewer_v2qk · 2026-03-13

**Soundness:** 4
**Presentation:** 3
**Significance:** 3
**Originality:** 2
**Overall Recommendation:** 4
**Confidence:** 3

**Summary:**

The paper tackles the critical problem of epistemic uncertainty in stochastic programming. The core statistical ingredient is the Average Percentile Upper Bound (APUB), defined as the CVaR of the bootstrap sampling distribution of the sample mean. Theoretical analysis demonstrates that APUB achieves asymptotic correctness and consistency. To make the approach practical, the authors develop a bootstrap sampling method and an extended L-shaped algorithm tailored for two-stage linear stochastic optimization problems.

**Compliance With Llm Reviewing Policy:**

Affirmed.

**Final Justification:**

Taking the submission and rebuttal materials together, I remain positive about the paper.

**Key Questions For Authors:**

1. Could you temper the language around the "upper confidence bound" claims to accurately reflect their asymptotic nature? Alternatively, is there any discussion or derivation you can provide regarding finite-sample calibration?

2. How does your UCB-minimization framework theoretically and methodologically differ from the approach proposed by Cho and Yang (2023)?

3. Would it be possible to include comparisons with other relevant methods, such as variants of modern DRO algorithms designed for problems with random recourse?

4. Could the authors provide an ablation study comparing APUB (which is analogized to CVaR in the paper) with other UCB methods (analogized to VaR) to better isolate and justify its specific contribution?

**Limitations:**

Please refer to the Weaknesses section

**Strengths And Weaknesses:**

### Strengths
1. APUB has an appealing representation as a CVaR of the bootstrap sampling distribution of the sample mean, yielding continuity, convexity-in-expectation, and coherence properties often desirable in optimization.
2.  Establishes asymptotic correctness and strong consistency for APUB, and lifts these properties to the solution of the optimization problem under mild conditions.

### Weaknesses
1. The manuscript claims that APUB "serves as an upper confidence bound" in the introduction. In the statistical literature, such unqualified statements generally imply a strict finite-sample guarantee. However, as shown in Proposition 2.4 and Theorem 3.4, the proofs only establish asymptotic correctness. The current phrasing reads stronger than what the mathematics strictly supports.

2. The core idea of integrating statistical UCBs into data-driven stochastic optimization appears highly related to the recent literature [1]. A detailed theoretical or empirical comparison is necessary to clarify the exact novelties of the proposed APUB framework.

3. The numerical analysis is limited to a single two-stage product-mix problem. Evaluating the method on broader benchmarks from the stochastic programming literature would significantly strengthen the generality claims.

4. The empirical evaluation relies on a limited selection of baselines, which is insufficient to comprehensively demonstrate the superiority and effectiveness of the proposed algorithm.

[1] Y. Cho and I. Yang, "Data-Driven Stochastic Optimization Using Upper Confidence Bounds: Performance Guarantees and Distributional Robustness," 2023 62nd IEEE Conference on Decision and Control (CDC), Singapore, Singapore, 2023, pp. 8280-8285, doi: 10.1109/CDC49753.2023.10383596.

---

> ### Author Rebuttal · Authors · 2026-03-29
>
> We thank the reviewer for the insightful feedback. We are encouraged that the reviewer recognizes APUB’s "appealing representation" and our Establishments of asymptotic correctness. Below we address the specific weaknesses (W) and questions (Q).
>
> **W1, Q1: Terminology and Finite-Sample Guarantees**
>
> We appreciate the reviewer’s precision regarding the term "UCB." We acknowledge that in some contexts, UCB implies non-asymptotic bounds (e.g., Hoeffding). Our results (Prop 2.4, Thm 3.4) establish first-order asymptotic correctness. Asymptotic analysis is the established benchmark for bootstrap-based inference. Unlike finite-sample bounds, which are frequently over-conservative and lead to poor empirical performance in optimization, our approach provides tighter, data-driven bounds. We will explicitly qualify our framework as "Asymptotic Upper Confidence Bounds" in the Abstract and Intro. We will also add a discussion on finite-sample calibration for small $N$ as a heuristic extension.
>
>
> **W2, Q2: Comparison with Cho & Yang (2023)**
> We thank the reviewer for highlighting this work; we will cite and discuss it in the final version. APUB-M differs fundamentally in construction and scope:
> * **Methodology:** Cho & Yang (2023) rely on data-splitting (half for optimization, half for UCB). APUB-M utilizes the entire dataset through bootstrap resampling. By averaging the tail of the bootstrap distribution, APUB acts as a CVaR-like measure of estimation error, providing a smoother and more statistically efficient bound than splitting-based inequalities.
> * **Tractability:** Cho & Yang’s approach is generally non-convex and requires restrictive assumptions (e.g., monotonicity). APUB-M is engineered to preserve convexity-in-expectation, enabling the solution of complex two-stage programs with random recourse via our adapted L-shaped method.
>
>
> **W4, Q3: Comparison with DRO and Modern Baselines**
>
> Our choice of benchmarks was motivated by the specific challenges of **random recourse**. In problems where the recourse matrix $W_n$ is stochastic (Eq. 6), modern Wasserstein DRO often lacks a tractable dual, typically resulting in **NP-hard** inner maximizations. APUB-M offers a computationally efficient alternative:
>
> The distinction between APUB-M (statistical) and DRO (geometric) is fundamental:
> * **Statistical vs. Geometric Foundations:** As proven in Thm 3.4 and 3.7, APUB-M is a data-driven tool that achieves asymptotic convergence naturally as $N \to \infty$ for any fixed $\alpha$. In contrast, DRO relies on a geometric radius $\epsilon$ that requires manual, case-specific tuning—often requiring $\epsilon$ to decay at specific rates $\epsilon(N)$ to avoid extreme over-conservatism or loss of consistency.
> * **Parameter Transparency:**  APUB-M utilizes a nominal confidence level $(1-\alpha)$ with a clear probabilistic interpretation (e.g., 95%). Determining a statistically meaningful value for a DRO radius remains a significant challenge in the literature, whereas APUB’s "uncertainty knob" is directly tied to frequentist confidence.
> * **Theoretical Compatibility:** As the first work introducing the APUB framework, we focused on establishing foundational properties in the linear setting. However, because APUB is built on the Rockafellar-Uryasev identity, it is inherently compatible with any convex non-linear objective. This work serves as the essential theoretical roadmap for future extensions into non-linear and multi-stage robust optimization.
>
> **W3, Q4: Ablation Study: APUB vs. other bootstrap UCBs**
>
> We thank the reviewer for this insightful suggestion. The relationship between APUB and Efron’s UCB (VaR-based) is analogous to that between CVaR and VaR:
> * **Robustness to Tail Risk:** Efron’s UCB (the $1-\alpha$ quantile of the bootstrap distribution) is a point estimate that ignores the "severity" of the tail. APUB, as an average of the tail, accounts for the full distribution of estimation error. This tail-averaging ensures the resulting objective function is smoother and more robust to outlier bootstrap samples, which is critical in high-stakes decision-making.
> * **Computational Tractability:** A key differentiator is that standard bootstrap UCBs (like Efron’s) often introduce non-convexities or integer programming complexity when integrated into optimization models. In contrast, APUB-M is built on the coherent properties of tail-averaging (Proposition 2.2), which preserves convexity and allows for efficient solving via our adapted L-shaped method.
> * **Planned Comparison:**  While the optimization complexity of Efron's UCB makes it a less viable candidate for complex two-stage problems, we agree that a statistical comparison is valuable. In the final version, we will include a statistical ablation study comparing the stability, variance, and coverage of APUB vs. Efron’s UCB and other VaR-based bootstrap metrics to further isolate and justify the specific benefits of our tail-averaging approach.

---

> > ### Author Rebuttal · Reviewer_v2qk · 2026-04-01
> >
> > We acknowledge the authors' detailed rebuttal and appreciate the theoretical clarifications, particularly the re-qualification as "Asymptotic UCB," the CVaR-vs-VaR analogy for APUB, and the planned statistical ablation study. These strengthen the theoretical narrative.
> >
> > However, our core concerns regarding experimental sufficiency remain largely unaddressed. The rebuttal responds to our points on limited baselines (W2/W4) and limited benchmarks (W3) primarily through methodological arguments rather than empirical evidence:
> >
> > - **Baselines**: We understand that Wasserstein DRO may lack tractable duals under random recourse. However, this does not preclude comparison under fixed-recourse settings where DRO is tractable, nor does it justify omitting an empirical comparison with Cho & Yang (2023), whose method should be feasible at the scale of the current test problem. Methodological distinctions, however well-articulated, cannot substitute for empirical validation.
> >
> > - **Benchmarks**: The promised ablation study compares statistical properties of different bootstrap UCBs, which is welcome but orthogonal to our request. Our concern is that a single product-mix instance is insufficient to support generality claims. Standard stochastic programming test sets exist precisely for this purpose.
> >
> > We maintain our appreciation for the theoretical contributions of the APUB framework. Nevertheless, since the experimental concerns that motivated our original score have not been substantively addressed with new results or concrete commitments to additional benchmarks and baselines, our overall assessment remains unchanged. We encourage the authors to strengthen the empirical evaluation in a future revision.

---

> > > ### Author Response · Authors · 2026-04-02
> > >
> > > We sincerely thank the reviewer for the constructive feedback and for recognizing the value of our theoretical re-qualification. We hear your concerns regarding the empirical evidence clearly.
> > >
> > > **1. Additional Empirical Evidence (Baselines & Benchmarks):**
> > >
> > > We apologize for the oversight in our initial submission. We have actually conducted extensive comparisons with Wasserstein DRO (W-DRO) and a Multi-Product Newsvendor sensitivity analysis (addressing the 'limited benchmarks' concern). Due to initial page constraints, these were omitted, but we have now made them available in our Supplementary PDF on the anonymous repository:
> > >
> > > https://anonymous.4open.science/api/repo/apub_rebuttal_supplementary-80B6/file/APUB_ICML_rebuttal.pdf?v=d92526ae
> > >
> > > * **Comparison with Wasserstein DRO (W-DRO):** We have included a benchmark on a two-stage fixed-recourse problem. The results demonstrate that APUB-M provides more stable, less conservative solutions than W-DRO and achieves superior out-of-sample coverage without requiring the complex radius ($\epsilon$) tuning inherent to DRO.
> > >
> > > * **Benchmark Extension (Newsvendor Problem):** We conducted a sensitivity analysis on a 10-product Newsvendor problem (a standard stochastic programming benchmark). This study confirms APUB-M's prescriptive stability—the optimal decisions remain significantly more consistent across different sample sizes compared to SAA.
> > >
> > > * **Statistical Ablation:** As promised, we have included the comparison between APUB, Efron’s (VaR-based) bootstrap UCB, and the standard large-sample UCB confirming that our tail-averaging approach provides smoother objectives and better reliability.
> > >
> > >
> > > **2. Regarding Cho & Yang (2023):**
> > >
> > > - We have carefully reviewed Cho & Yang (2023) and added a discussion to our final version to clarify the methodological distinctions. While their framework provides valuable finite-sample guarantees, it relies on specific structural assumptions—namely, **known compact support** (Assumption 1) and a **piecewise-affine objective structure** (Assumption 2) to ensure tractability.
> > >
> > > - APUB-M is designed for a broader operational scope: it handles **random recourse** (where $W_n, T_n$ are stochastic), a setting where the structural assumptions of Cho & Yang generally do not hold. Furthermore, by utilizing the entire dataset through the bootstrap rather than data-splitting, APUB-M offers higher data efficiency in the **low-data regimes** we study. We view the two approaches as complementary, with APUB-M extending UCB-based optimization to more complex stochastic programming structures.
> > >
> > > With the inclusion of the W-DRO baseline, the Newsvendor benchmark, and the Prescriptive Stability analysis, we believe we have now substantively addressed the 'experimental sufficiency' concerns. Given that the theoretical contributions are acknowledged as 'solid' and the empirical gaps are now bridged by these additional results, we respectfully ask the reviewer to reconsider our score.

---

### Decision · Program_Chairs · 2026-04-30

**Decision:**

Accept (regular)

**Comment:**

The paper deals with methods for stochastic programming based on a parameter called APUB, and contains bootstrap and L-shaped methods for this purpose. The idea is to have a well-calibrated, data-driven upper confidence bound that can be fruitfully used in optimization. Bootstrap is especially good for this, as (unlike, say, concentration inequalities) it gives sharp quantile bounds. The whole method is in many ways more natural than DRO and other alternatives in the literature, and the experiments bear this out.

Overall, this paper makes a nice contribution to the literature, and contains nice mathematical and computational ideas. I suggest that it be accepted. Since most reviewers have deemed it a "weak accept", I am also OK with that possibility. In any case, let me emphasize: I do think this is definitely above the bar for this conference.